# Wind Conditions for Snow Cornice Formation in a Wind Tunnel

Hongxiang Yu[1,3], Guang Li[2,3], Benjamin Walter[4], Michael Lehning[3,4], Jie Zhang[1], and Ning Huang[1]

[1]College of Civil engineering and Mechanics, Lanzhou University, Lanzhou, 730000, China
[2]College of Atmospheric Science, Lanzhou University, Lanzhou, 730000, China
[3]College of Architecture Civil and Environmental Engineering, Ecole Polytechnique Federal de Lausanne, Lausanne, 1015, Switzerland
[4]WSL Institute for Snow and Avalanche Research SLF, Davos, 7260, Switzerland

**Correspondence:** Ning Huang (huangn@lzu.edu.cn), Michael Lehning (lehning@slf.ch)

**Abstract.** Snow cornices growing on the leeward side of mountain ridges are common in alpine and polar regions during snow seasons. These structures may crack and fall, leading to an increase in avalanche danger. Although cornice formation has been observed in wind tunnel tests and the field, knowledge gaps still exist regarding the formation mechanism. This is particularly true with respect to wind conditions which favor cornice formation. To characterize the wind effects as the main factor for cornice growth, we carried out ring wind tunnel (RWT) experiments in a cold laboratory under various wind conditions. We quantitatively investigated the growth rate of the cornice in the horizontal and vertical direction, as well as the airborne particle concentration. The results show that cornices only appear under a moderate wind speed range (1–2 times the threshold wind speed). The cornice growth rates in length and thickness are mainly determined by the combined effects of mass accumulation and erosion. The lower limit wind speed for cornice growth is approximately equal to the threshold wind speed for snow transport. The upper limit of wind speed is when the erosion rate is higher than the deposition rate. The length growth rate of the cornices reaches a maximum for wind speeds approximately 40% higher than the threshold wind speed. Moreover, a conceptual model for interpreting the cornice accretion mechanism is proposed based on the mass conservation and the results of the RWT experiments. The estimated suitable wind condition for cornice growth and formation are in good agreement with field observations in Gruvefjellet, Svalbard. Based on the physics of drifting snow, our results provide new insights into snow cornice formation and improve understanding of cornice processes that can influence avalanche activity. The experimental results and the conceptual model can be used in future snow cornice simulation and prediction work for cornice-induced avalanches.

**NOTATION**

| Symbol | Definition and units |
|--------|---------------------|
| $A_0$ | Area of one pixel [m$^2$] |
| $d_\mathrm{p}$ | Particle diameter [m] |
| $dt$ | Time interval [s] |
| $E$ | Collection efficiency [%] |
| $f_\mathrm{l}$ | Horizontal collection coefficient [m s$^{-1}$] |
| $g$ | Gravitational acceleration [= 9.81 m s$^{-2}$] |
| $H$ | Cornice thickness [m] |
| $h_\mathrm{d}$ | Deposition rate in thickness [m s$^{-1}$] |
| $\bar{h}_\mathrm{d}$ | Averaged deposition rate in thickness [m s$^{-1}$] |
| $h_\mathrm{e}$ | Erosion rate in thickness [m s$^{-1}$] |
| $\bar{h}_\mathrm{e}$ | Averaged erosion rate in thickness [m s$^{-1}$] |
| $h_\mathrm{g}$ | Growth rate in thickness [m s$^{-1}$] |
| $\bar{h}_\mathrm{g}$ | Averaged growth rate in thickness [m s$^{-1}$] |
| $\Delta h$ | Thickness of newly formed snow layer [m] |
| $K_\mathrm{d}$ | Deposition coefficient [m s$^{-1}$] |
| $k$ | Shape factor |
| $\kappa$ | Von Kármán constant [= 0.4] |
| $L$ | Cornice length [m] |
| $l_\mathrm{d}$ | Deposition rate in length [m s$^{-1}$] |
| $\bar{l}_\mathrm{d}$ | Averaged deposition rate in length [m s$^{-1}$] |
| $l_\mathrm{e}$ | Erosion rate in length [m s$^{-1}$] |
| $\bar{l}_\mathrm{e}$ | Averaged erosion rate in length [m s$^{-1}$] |
| $l_\mathrm{g}$ | Growth rate in length [m s$^{-1}$] |
| $\bar{l}_\mathrm{g}$ | Averaged growth rate in length [m s$^{-1}$] |
| $l_\mathrm{g}^\mathrm{f}$ | Growth rate in length the field [m s$^{-1}$] |
| $\Delta l$ | Moving distance of a newly formed snow layer [m] |
| $M_\mathrm{e}$ | Erosion rate of mass [kg m$^{-2}$ s$^{-1}$] |
| $m_\mathrm{p}$ | One particle mass [kg] |
| $p$ | Depth of field [m] |
| $Q$ | Transport rate [kg m$^{-1}$ s$^{-1}$] |
| $q_\mathrm{p}$ | Particle mass flux [kg m$^{-2}$ s$^{-1}$] |

**NOTATION**

| | |
|---|---|
| $S_0$ | Area of window $\Omega$ [m$^2$] |
| $S_c$ | Cornice area [m$^2$ s$^{-1}$] |
| $t$ | Time [s] |
| $T$ | Sampling time of the scanner images [min] |
| $u$ | Wind speed [m s$^{-1}$] |
| $\tilde{u}$ | Non-dimensional wind speed |
| $\bar{u}$ | Daily averaged wind speed [m s$^{-1}$] |
| $u_f$ | Field wind speed [m s$^{-1}$] |
| $u_p$ | Particle velocity [m s$^{-1}$] |
| $u_t$ | Threshold wind speed [m s$^{-1}$] |
| $u_*$ | Friction velocity [m s$^{-1}$] |
| $u_{*t}$ | Threshold friction velocity [m s$^{-1}$] |
| $z$ | Height above snow surface [m] |
| $z_0$ | Roughness length [m] |
| $z_f$ | Height of wind speed sensor in field [m] |
| $\rho_a$ | Air density [kg m$^{-3}$] |
| $\rho_c$ | Cornice density [kg m$^{-3}$] |
| $\rho_i$ | Ice density [kg m$^{-3}$] |
| $\eta_{ae}$ | Aerodynamic entrainment coefficient [grains N$^{-1}$ s$^{-1}$] |
| $\phi_p$ | Mass concentration of particles in the air [kg m$^{-3}$] |
| $\lambda$ | Scale factor |

# 1 Introduction

Snow cornices are leeward-growing masses of snow overhanging and extending horizontally beyond the edge, usually appearing on the ridgeline of steep mountains (Seligman et al., 1936). Some cornices deform, detach, and eventually fall off, which induces cornice fall avalanches or slope erosion, and leads to a redistribution of the snow cover below (Wahl et al., 2009). For example, cornice fall avalanches accounted for 45.2% of all 423 snow avalanches observed in the Longyearbyen area, central Svalbard, from 2006 to 2009. They triggered slab avalanches and loose snow avalanches as secondary avalanches on the slope below (Eckerstorfer and Christiansen, 2011). Cornice fall avalanches cause potential threats to local infrastructure and human lives.

Although understanding the initial evolution of cornices is a foundation for predicting and treating cornice fall avalanches, only a few studies have paid attention to the initial accretion process, especially to the horizontal extension forming the main part of the snow mass overhanging the edge of a mountain crest. Previous research has observed that cornices grow under moderate wind speeds. However, gaps remain regarding a general rule on suitable wind conditions for cornice growth. Montagne et al. (1968) measured a wind speed range between 7 and 15 m s$^{-1}$ (at 0.35 m height) for cornice formation using a hand anemometer. Naito and Kobayashi (1986) identified wind speed between 4 to 8 m s$^{-1}$ as suitable for cornice formation at 1 m above the snow surface in the field and at the center (0.5 m height) in the wind tunnel. McClung and Schaerer (2006) estimated that the threshold wind speed for cornice growth and formation is about 5 to 10 m s$^{-1}$ (at 10 m height), which is the threshold wind speed for loose snow transport, and scouring happens when the wind speed exceeds 25 m s$^{-1}$. Vogel et al. (2012) determined that cornice accretion occurs during periods with the average wind speed of 12 m s$^{-1}$, and scour when the wind speed exceeds 30 m s$^{-1}$ (at 2.8 m height). Hancock et al. (2020) used wind speed of 5 m s$^{-1}$ (at 10 m height) as a conservative lower threshold for cornice accretion. However, to our best knowledge, these discrepancies and the conditions under which certain wind speed ranges apply have not been investigated. Furthermore, how the wind erosion threshold and thus cornice formation depend on the snow micro-structure properties like the snow grain size and dendricity has not been fully understood.

Indirect evidence was presented by van Herwijnen and Fierz (2014) that snow cornices only grow under moderate to strong winds, during or soon after a snowfall. The cornice width from observation is in remarkable agreement with the wind drift index calculated by the snow cover model SNOWPACK (Lehning and Fierz, 2008), which indicates that snow mass transport plays an important role in cornice formation. However, cornices often grow through relatively discrete events in the field (Vogel et al., 2012; van Herwijnen and Fierz, 2014; Naito and Kobayashi, 1986; Hancock et al., 2020). Daily observations therefore only incompletely characterize cornice growth conditions. Due to the compromise of these field observations, continuous observations on individual cornice accretion and failure events are hard to achieve (Hancock et al., 2020). Specifically, measuring the horizontal growth of snow cornice (Vogel et al., 2012) and recording dynamic details of snow mass transport simultaneously is hard to achieve. There are only two laboratory experiments on cornice formation (Naruse et al., 1985; Naito and Kobayashi, 1986). Naito and Kobayashi (1986) carried out experiments both in the wind tunnel and in the field, observing the process of snow cornice growth. They described snow cornice formation as a process in which drifting snow particles adhere one after another at the leeward edge. The formed thin snow slab elongating leeward then hangs down under its weight.

Their results show suitable conditions for cornice growth include the air temperatures of -20 to 0°C, wind speeds 4 to 8 m s⁻¹,

and irregular dendritic-shaped snowflakes with larger contact surface. However, quantitative descriptions of this process have not been reported, and further quantitative analysis of experiments have not been carried out. Moreover, the locations such as cornice-like deposition at the ridge are well predicted in the numerical simulation using Alpine3D (Lehning et al., 2006) and ARPS (Mott et al., 2010), but the cornice shape cannot be represented.

Therefore, wind tunnel experiments under controlled environmental conditions and quantitative descriptions of the individual

cornice formation process as a pathway to improve the understanding of cornice dynamics in the field, particularly on the wind effects on cornice formation, are essential. In this work, wind tunnel experiments of snow cornice evolution on the edge of a small-scaled mountain ridge model carried out in a cold laboratory at the WSL Institute for Snow and Avalanche Research (SLF) are presented. A quantitative analysis of the effect of wind conditions on snow cornice formation is presented. Section 2 presents the experimental setup in the cold laboratory and the post-processing method for the cornice images. General features

of the snow cornice observed in the experiment under variable wind conditions are shown in Section 3. Finally, a conceptual model for the growth rates of snow cornices based on mass conservation is proposed in Section 4. Its application for field observations is discussed. Section 5 summarizes the conclusions and outlook.

## 2   Methods

### 2.1   Experimental Setup

The experiments were carried out in a cold laboratory of the SLF in Davos, Switzerland, where the room temperature can be controlled from −25 to 0 °C. An obround, closed-circuit wind tunnel built by Sommer et al. (2017, 2018) was used to perform the investigations. During the experiments, the room temperature of the cold laboratory was set to be −5 °C. At this temperature, the cohesion of snow particles is significantly enhanced compared with colder temperatures (Tobias et al., 2022).

The schematic diagram of the experimental setup is shown in Fig. 1. The ring wind tunnel (RWT) contains two straight

sections (length = 1 m, marked as S1 and S2) and two half-circle sections (outer diameter = 0.6 m, marked as H1 and H2). Its cross-section area is 0.2 m (width) × 0.5 m (height). An electric motor with rotor blades installed inside the middle of H1 creates the wind flow with a wind speed range of 0–8 m s⁻¹. A sieve is installed at S1, where the tunnel has an upward open window to supply snow particles. Sensors monitoring the air conditions are installed at the inlet of S2. The details of the sensors are listed in Table1. The ridge model in S2 with the fixed size and place is built with compacted snow each time before the

experiment. The size of the ridge model was set as 0.125 m in height and with a 0.1 m flat section. The slope angle relative to the horizontal direction is 36 °. To record the growth of the cornice using shadowgraphy imaging, we placed a CMOS Camera with a spatial resolution of 2048 × 2048 pixels to zoom on the edge of the ridge. We placed a LED lamp on the opposite side for illumination.

Fresh snow particles made with a snowmaker developed at SLF (Schleef et al., 2014) were used for feeding the flow through

the sieve. When using the snowmaker, the room temperature was set to −20 °C, and the water inside the snowmaker reservoir was set to 30 °C. The obtained fresh snow is a mixture of dendritic crystals and hollow columns. The average diameter was

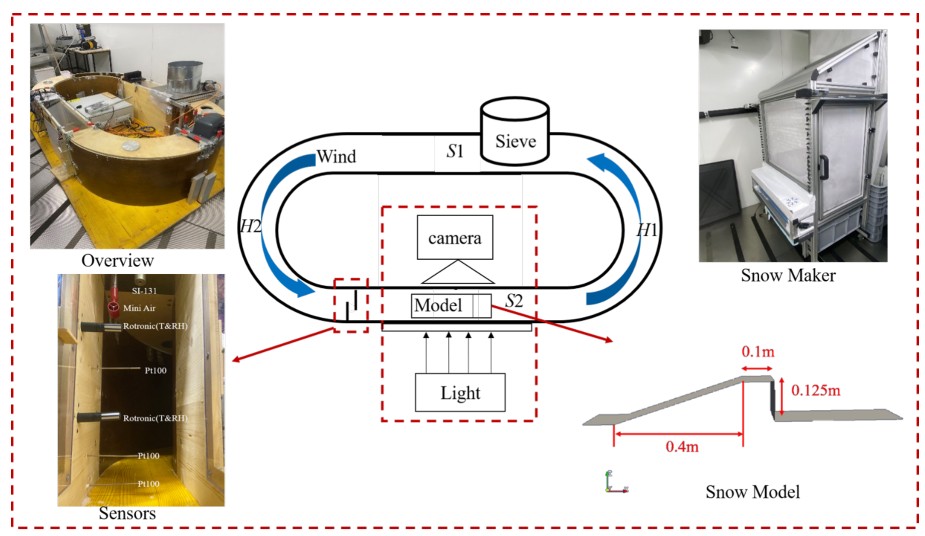

**Figure 1.** Schematic diagram of the closed-circuit tunnel experimental system in the cold lab. The insets are the pictures of the RWT, Snow Maker , sensors and the snow model set up inside the RWT.

**Table 1.** Instruments, variables, and data acquisition interval.

| Instrument | Instrument model | Variables | Time interval (s) |
|---|---|---|---|
| CMOS Camera | LP285-40.5 | Images | 0.02 |
| Wind Speed Tester | Mini Air | $u$ (m s$^{-1}$) | 0.2 |
| Snow Temperature | Pt100 | $T$ (K) | 1 |
| Snow Surface Temperature | SI-131 | $T_s$ (K) | 1 |
| Air Temperature and RH | Rotronic | $T$ (K) and RH (%) | 1 |

about 300–500 $\mu$m, estimated by a grid plate and an amplifying lens. The specific surface area (SSA) was about 40–60 mm$^{-1}$ for the snow that was stored for a few days up to a week (Schleef et al., 2014). A constant seeding rate is applied for all experimental tests, and the wind tunnel is cleaned up before each test. Impact threshold wind speed in the experiment is

90 determined 1) by increasing the wind speed from zero until saltating particles can be observed; 2) by decreasing the wind speed slowly until snow saltation is not visible anymore. The average wind speed at these two times is considered the impact threshold wind speed (Walter et al., 2014). The average impact threshold wind speed was 3.2 m s$^{-1}$ at the height of the mini-air wind sensor. Thus, seven target wind speed conditions (from 3.0 to 6.0 m s$^{-1}$ by steps of 0.5 m s$^{-1}$) were set for the experiments. Once the propeller starts to rotate, the wind speed increases until it reaches the target value. The propeller angular velocity is

95 adjusted throughout the experiment to keep the wind speed constant.

## 2.2 Image Processing

The CMOS camera recorded 50 images with a frequency of 10 Hz in the burst mode, and the pause between two bursts was 5 s. Thus, 50 continuous frames in 5 s as one set was obtained, which could be used to estimate the cornice growth rate and transport mass flux instantaneously or on average. The first image, in which only the ridge model was visible without snow particles moving across, was set as a background image, as shown in Fig. 2a. For a set of 50 frames, the images were subtracted from the background image (only with the ridge model) and transformed to binary format (where the grayscale value of pixels with snow is 1, and without snow is 0), as shown in Fig. 2c-e. The cornice length $L$ and cornice thickness $H$ are calculated based on the binary images (Fig. 2c-d). To avoid a wrong interpretation (as erosion or deposition) of the shape effect of bending, we used the thickness of accumulation mass on the flat as the indicator of vertical accumulation/erosion of the cornice in the following analysis.

The instantaneous cornice growth or erosion rate in thickness $h_{g/e}$ and in length $l_{g/e}$ are then calculated as the difference of two adjacent frames divided by the time difference between two images $\Delta t$:

$$h_{g/e} = \frac{\Delta H}{\Delta t} \tag{1}$$

$$l_{g/e} = \frac{\Delta L}{\Delta t} \tag{2}$$

The averaged deposition rates in length $\bar{l}_d$ and in thickness $\bar{h}_d$ are calculated as the sum of the growth rate and the absolute value of erosion rate:

$$\bar{h}_d = \bar{h}_g + |\bar{h}_e| \tag{3}$$

$$\bar{l}_d = \bar{l}_g + |\bar{l}_e| \tag{4}$$

A window $\Omega$ with an area of 1 cm $\times$ 1 cm slightly above the snow cornice is chosen to calculate the mean mass concentration of particles in the air as shown in Fig. 2e. Ignoring the overlapping particles, we calculate the total volume of snow particles in $\Omega$ as the orthographic projection area of snow particles multiplied by its average diameter. Thus, the mass concentration $\phi_p$ can be estimated as:

$$\phi_p = \frac{\rho_i \bar{d}_p \Sigma_\Omega g_j A_0}{S_0 \times p} \tag{5}$$

where $\rho_i$ is the ice density, $d_p$ is the averaged diameter, $g_j$ is the binary value of the $j^{th}$ pixel in window $\Omega$, $A_0 = \frac{7.7 \times 7.7}{2048 \times 2048}$ cm$^2$ is the area of a pixel, $S_0 = 1$ cm$^2$ is the area of the window $\Omega$, $p = 3.5$ cm is the depth of field where particles can be detected in this width range (Crivelli et al., 2016). The transport mass flux $q_p$ can be estimated using:

$$q_p(z) = \phi_p(z) u_p \tag{6}$$

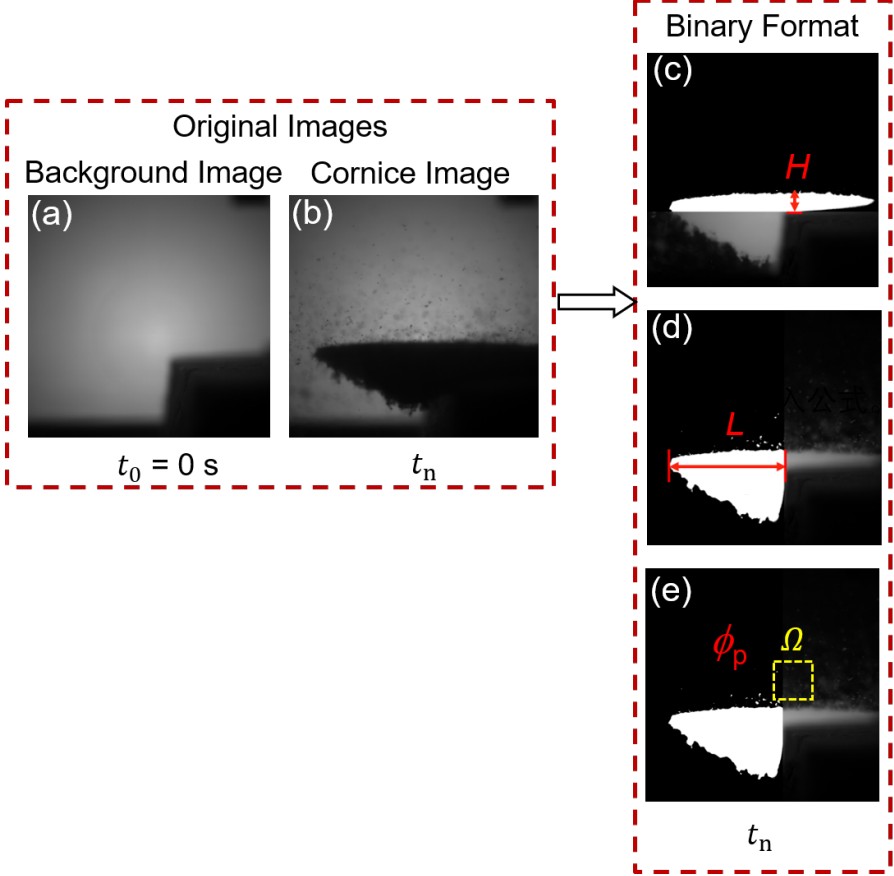

**Figure 2.** Post-Processing images using the grayscale method. Raw images of background (a) and cornice (b). The binary format of images with information of thickness accumulation $H$ (c), length growth $L$ (d), and airborne snow particles mass concentration $\phi_p$ captured in window $\Omega$ (e).

The transport mass flux profile can be described by an exponential law (Nishimura and Nemoto, 2005; Sugiura et al., 1998):

$$q_p(z) = Ae^{-R_0 z} \tag{7}$$

where $A$ and $R_0$ are empirical coefficients that change with wind speed. The transport rate $Q$ can be obtained by integrating the mass flux profiles over height:

$$Q = \int_0^\infty q_p(z)dz = \int_0^\infty Ae^{-R_0 z}dz = -\frac{A}{R_0}e^{-R_0 z}|_{z=0}^{z=\infty} = \frac{A}{R_0} \tag{8}$$

To quantify the exchange of snow between the mass flux and the cornice, we defined the relative mass flux collection efficiency as:

$$E = \frac{S_c \times \rho_c}{Q} \times 100\% \tag{9}$$

where $S_c = \frac{dA_c}{dt}$ is the growth rate of the cornice projected area $A_c$, and $\rho_c = 147$ kg m$^{-3}$ is the average snow density of the cornice as measured during the experiments. This value is close to the fresh snow and lower than that in the field of $\sim 300$ kg m$^{-3}$ (Naruse et al., 1985), which might be related to the long-term compaction of the snowpack in the field.

## 3   Results

### 3.1   General observations on snow cornice formation

By post-processing the high-speed camera images from the experiments, the profiles of the snow cornice are obtained as shown in Fig. 3a. The time series of the cornice length $L$, thickness $H$, and the mass concentration of airborne snow $\phi_p$ are then estimated as shown in Fig. 3b. Here, we use the case of wind speed $u = 4$ m s$^{-1}$ as an example to present the cornice growth process. As is shown in Fig. 3b, the cornice size information associated with wind speed and particle mass concentration are presented. The wind speed increased from 0 to 4 m s$^{-1}$ in about 210 s and was then kept stable during the cornice formation process. The particle mass concentration started to increase at $t = 176$ s (marked in black dash line: $u = u_t$) and reached a stable value at $t = 250$ s. The cornice started to grow when the wind speed exceeded the threshold. The growth rate was not stable at first because the initial growth of cornice is in intermittent drifting snow when the aerodynamic entrainment is still dominant in the initial stage of drifting snow (Li et al., 2018). The linear length growth stage is when the wind speed and mass concentration values arrive stable.

During cornice accretion, there are two stages for the growth of the cornice. In the first stage, a few particles stop on edge and compose a 0.011 m small and thin slab that forms leeward from the ridge model's edge. The shape profile of this slab is shown as from $t_1$ to $t_3$ in Fig. 3a. In the second stage (from $t = 320$ s in Fig. 3b), the cornice thickness grows simultaneously with the length. With more layers overlapping on the surface, the cornice starts slightly bending down.

When the cornice length reaches the boundary of the image, we stopped seeding and erosion then reduces the thickness of the snow cornice. The downward bending continues (outlines from $t_7$ to $t_8$ in Fig. 3a and $t = 430$–$440$ s in Fig. 3b). During this period, aerodynamic entrainment dominates the erosion process. As is shown in Fig. 3b, the mass flux markedly decreases as the aerodynamic entrainment is inhibited by the surface morphology formed during the redistribution of the snow deposition in the RWT.

### 3.2   Mass flux and collection efficiency

Since the magnitude of drifting snow is critical for the vertical and horizontal cornice growth rates, the mass transport rates were calculated for the different experiments and analyzed in terms of mass exchange between the cornice and the saltation

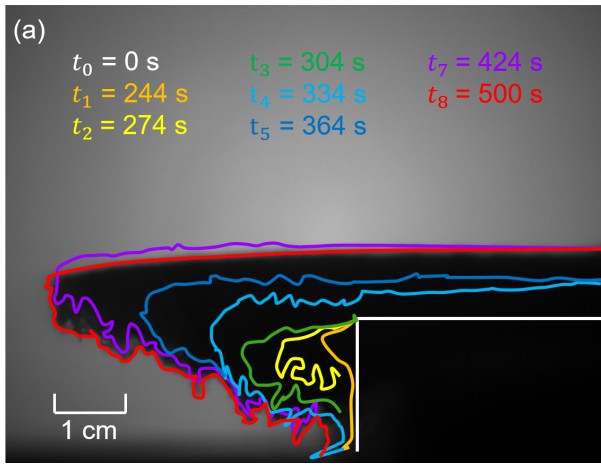

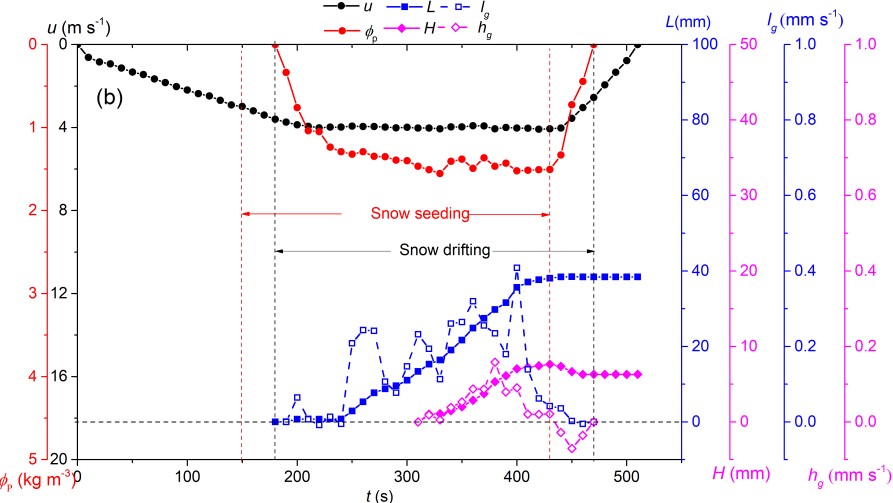

**Figure 3.** (a) Cornice profiles in the growth process. (b) Variation of cornice length (blue squares), thickness (pink squares), length growth rate (blue hollow squares), cornice thickness growth rate (pink hollow square), wind speed (black circles), and particle mass concentration in the air (red circles).

layer. The mass flux variation with height over snow cornices can be estimated by multiple windows $\Omega$ that are continuously distributed in height, as is shown in Fig. 4. The mass flux exponentially decreases with the increasing height in each wind condition, and its value increases with the wind speed, which is consistent with previous results (Takeuchi, 1980; Lehning et al., 2002; Kosugi et al., 2008; Lü et al., 2012; Crivelli et al., 2016; Melo et al., 2022).

By fitting Eq. (7) using the estimated mass flux from the shadow images, we obtain $A$ and $R_0$ for different wind speeds, as summarized in Table 2. Their fitted functions are: $A = -2092 + 1840u - 596u^2 + 84u^3 - 4u^4$ and $R_0 = -285.95 + 118.29u$. As is

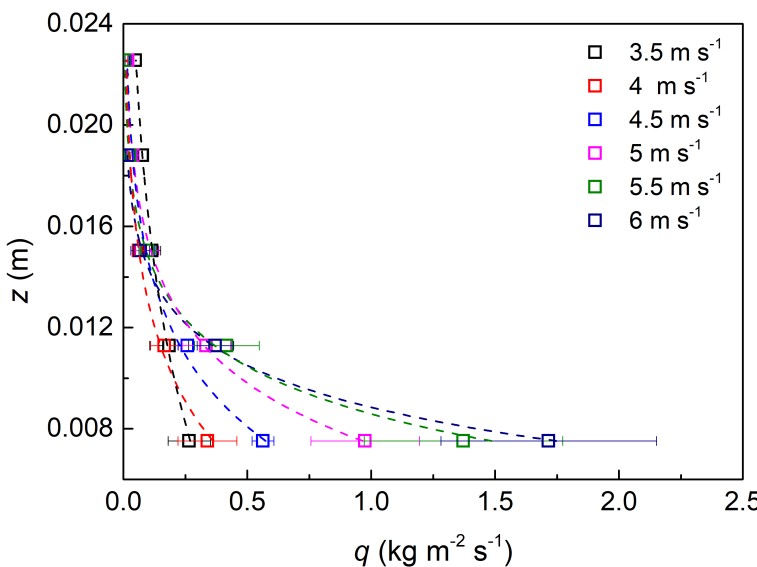

**Figure 4.** Mass flux variation with height under different wind conditions (3.5 - 6 m s$^{-1}$). The dashed lines are exponentially fitted.

**Table 2.** Coefficients of $A$ and $R_0$ for different wind speeds $u$.

| Wind Speed $u$ (m s$^{-1}$) | $A$ | $R_0$ |
|---|---|---|
| 3.5 | 0.62 | 109.42 |
| 4 | 2.09 | 218.5 |
| 4.5 | 3.63 | 245.32 |
| 5 | 8.44 | 288.16 |
| 5.5 | 24.05 | 370.31 |
| 6 | 40.72 | 418.78 |

shown in Fig. 4, the transport mass flux profile can be described by an exponential law (Nishimura and Nemoto, 2005; Sugiura et al., 1998).

A non-dimensional wind speed $\tilde{u} = \frac{u}{u_t}$ is defined here to compare with the experimental results of Naito and Kobayashi (1986). In this definition, $u_t$ is the threshold wind speed which can be considered as the lower limit wind speed value for cornice growth. As is shown in Fig. 5, the mass collection efficiency in both, N&K86 and our experiment decreases with the increasing wind speed and the corresponding drift rate. Our measured values for the collection efficiency are in the same order of magnitude as in N&K86. Due to the limited data and unpublished details in the study of N&K86, we could not make a deeper comparison.

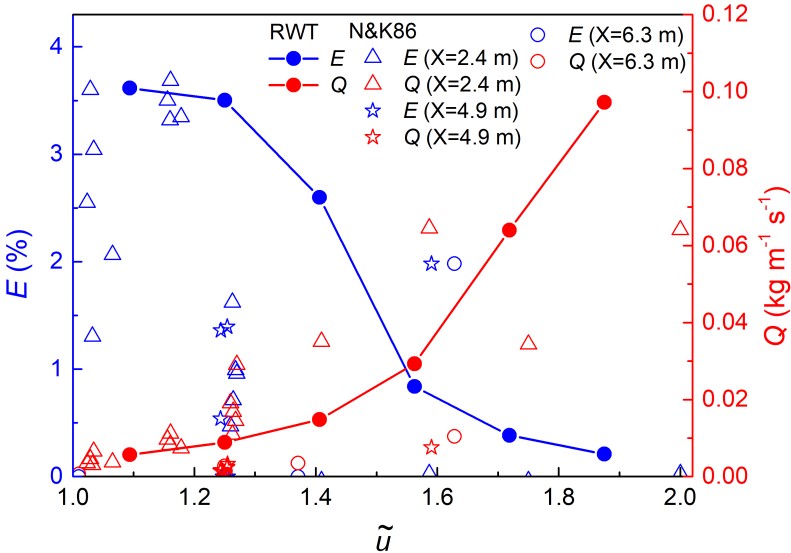

**Figure 5.** Collection efficiency $E$ (in blue) and snow transport rate $Q$ (in red) under different non-dimensional wind speeds $\tilde{u}$. X represents the distance from the snow particle feeding point to the mass collection pits where the cornice grows. Lines are for ring wind-tunnel experiments, and hollow scatters are for N&K86. N&K86 represents the experiment results of Naito and Kobayashi (1986)

The collection efficiency cannot directly reflect the cornice growth characteristics because it represents the proportion of snow particles passing through the edge and stopping by. This value only reflects the effective contribution of the drifting snow to the snow cornice formation under different wind conditions. Thus, to characterize the growth rate of cornice, it is necessary to analyze the absolute amount of accumulated particles as a function of time and wind speed which is introduced in Section 3.3.

### 3.3 The suitable wind speed range for cornice formation

Cornice formation was tested with wind speeds from 3.0 to 6.0 m s$^{-1}$ using 0.5 m s$^{-1}$ increments. For each wind condition, the averaged cornice growth rates in length and thickness are: $\bar{l}_g$ and $\bar{h}_g$ (with seeding), and erosion rates: $\bar{l}_e$ and $\bar{h}_e$ (without seeding) are obtained by taking the average of the growth/erosion rates during the linear increase/decrease of the length/height, as is shown in the Fig. 3. Thus, the averaged deposition rates in length $\bar{l}_d$ and thickness $\bar{h}_d$ can be calculated by the Eq. (3-4).

As is shown in Fig. 6, there is no cornice formation for wind speed lower than the threshold wind speed because of a missing saltation layer and snow transport. The extension line of the deposition rate in length tends to zero around the threshold wind speed for snow transport. Thus, we can conclude that the lower limit wind speed for cornice accretion is close to the threshold wind speed for snow transportation, which is consistent with the field study (McClung and Schaerer, 2006; Hancock et al., 2020).

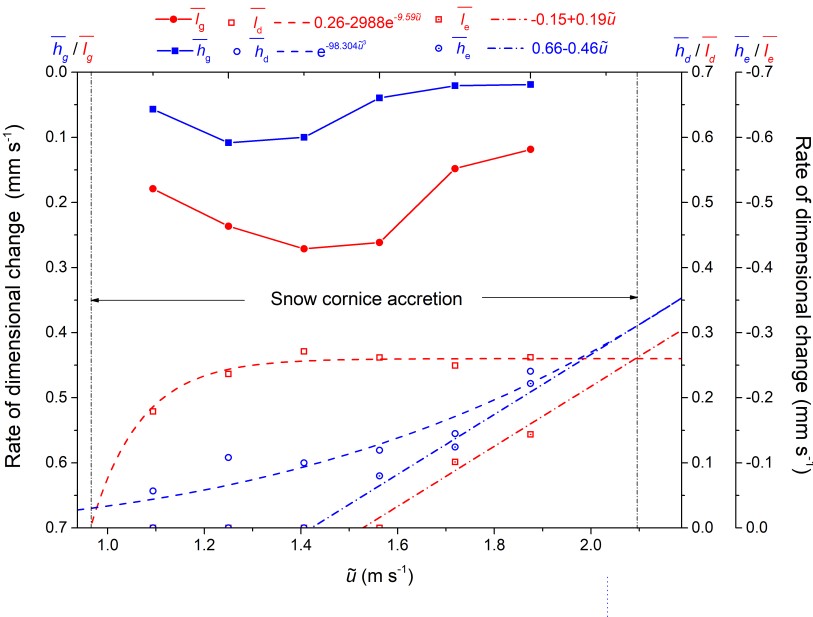

**Figure 6.** Averaged growth rates, erosion rates, and deposition rates in length and thickness under different wind conditions. The fitted functions are plotted in dashed lines.

The averaged cornice length growth rate $\bar{l}_g$ (equal to $\bar{l}_d - \bar{l}_e$) reaches its maximum when the wind speed is approximately 40% higher than the threshold wind speed. The averaged erosion rates in length and thickness approximately linearly increase with the wind speed. In the cornice growing process, the averaged length growth rate ($\bar{l}_g$) is higher than the averaged thickness growth rate ($\bar{h}_g$) at all wind speed conditions. The erosion in length takes place later than in thickness, and the averaged thickness erosion rate is always approximately 30% higher than the length erosion rate ($\bar{h}_e = 1.3\bar{l}_e$).

The averaged deposition rate in length $\bar{l}_d$ increases rapidly at first and stabilizes with the wind speed, while the averaged erosion rate in length $\bar{l}_e$ linearly increases with the wind speed. The values of $\bar{l}_d$ and $\bar{l}_e$ arrive equivalent at the wind condition of about 6.5 m s$^{-1}$, at which point the mass of the accumulation and the erosion is balanced. Thus, the upper limit wind speed of snow cornice formation in our case is 6.5 m s$^{-1}$, which is 2 times of threshold wind speed.

Overall, the cornice growth process has two stages: In the first stage, a thin slab grows and overhangs at the edge. In the second stage, cornice thickness and length both increase simultaneously. The collection efficiency, reflecting the effective contribution of the drifting snow to the snow cornice formation, cannot directly reflect the cornice growth characteristics. Instead, the deposition rates, the erosion rates, and the growth rates both in length and thickness were analyzed separately for all wind conditions. From the results we can conclude that in all wind conditions, the cornice starts to grow when the wind speed exceeds the threshold value, and starts to scour when the erosion rate is higher than the deposition rate. The cornice only grows at a moderate wind speed range (1–2 $u_t$). The length erosion rate of the cornice is typically 30% lower relative to the

thickness erosion rate. The presented framework for characterizing cornice accretion may provide a basis for future field and laboratory studies under different conditions.

## 4 Discussion

From the experimental results, we can conclude that cornice growth is a process of mass accumulation overgrowing the ridge
under the action of wind force, accompanied by bending and erosion. The growth process of a snow cornice has two stages, which can be described with a schematic shown in Fig. 7. The first stage can be assumed as a formation of a 1–2 particle diameters thick snow slab composed of particles sticking horizontally at the edge (see Fig. 3a and b). The first process is mainly determined by the spatial variation of the mass transport rate along the flow direction. The second stage can be assumed to be a repeated process of length growth – thickness growth. The length growth is considered as a horizontal creeping of
the newly formed snow layer, driven by the drifting snow. The thickness growth is considered as a comprehensive result of particle deposition and erosion at the edge. Thus, the second growth process is mainly dependent on the wind speed, the non-dimensional, spatial variation of mass concentration, and the particle interaction force.

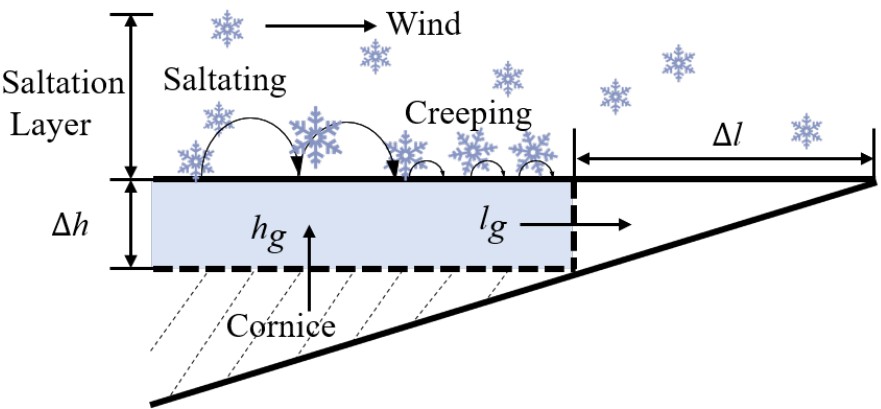

**Figure 7.** Schematic of snow cornice growth.

### 4.1 A conceptual model for cornice formation

In this Section, we analyze the snow cornice as the shaded area shown in Fig. 7. Based on the law of mass conservation,
the cornice thickness growth rate $h_g$ can be calculated as the difference between the deposition rate in thickness $h_d$ and the thickness erosion rate $h_e$. In which, the deposition rate in thickness $h_d = K_d \frac{\phi_p}{\rho_c}$ is calculated by the deposition rate of mass on the surface per unit of time. Thus, the thickness growth rate $h_g$ can be written as:

$$h_g = K_d \frac{\phi_p}{\rho_c} - h_e \tag{10}$$

where $K_d$ is the deposition coefficient.

The cornice length growth is considered as the forward creeping of the surface layer, which is driven by the drifting snow saltation. The cornice length growth rate $l_g$ can be estimated as the difference between the deposition rate in length $l_d$ and the length erosion rate $l_e$. In which $l_d$ is considered as the horizontal moving distance $\Delta l$ of a newly formed snow layer (blue area in Fig. 7) with 1-2 particle diameters in thickness $\Delta h$ per unit time. The deposition rate in length $l_d$ is related to the mass transport rate $Q$ and the non-dimensional horizontal collection coefficient $f_l$:

$$l_g = \frac{Q f_l}{\rho_c d_p} - l_e \tag{11}$$

     As we already measured the cornice thickness growth rate $h_g$, thickness erosion rate $h_e$, the cornice length growth rate $l_g$, length erosion rate $l_e$, the air mass concentration $\phi_p$, and the mass transport rate $Q$, the deposition coefficient and the horizontal collection rate can be estimated as $K_d = \frac{(h_g + h_e)\rho_c}{\phi_p}$ and $f_l = \frac{(l_g + l_e)\rho_c d_p}{Q}$, which exponentially decrease with the wind speed, as shown in Fig. 8. Although deposition and collection coefficients are largest at the lowest wind speeds, the total mass flux is still

too small to result in a significant length/height growth. At around 40% above the threshold, $Q f_l$ are maximum resulting in the strongest length growth rates $l_g$. The net height and length growth rate from Eq. 10 and Eq. 11 together with the deposition collection rates from Fig. 8 can be used to simulate cornice accretion for a wide range of atmospheric conditions.

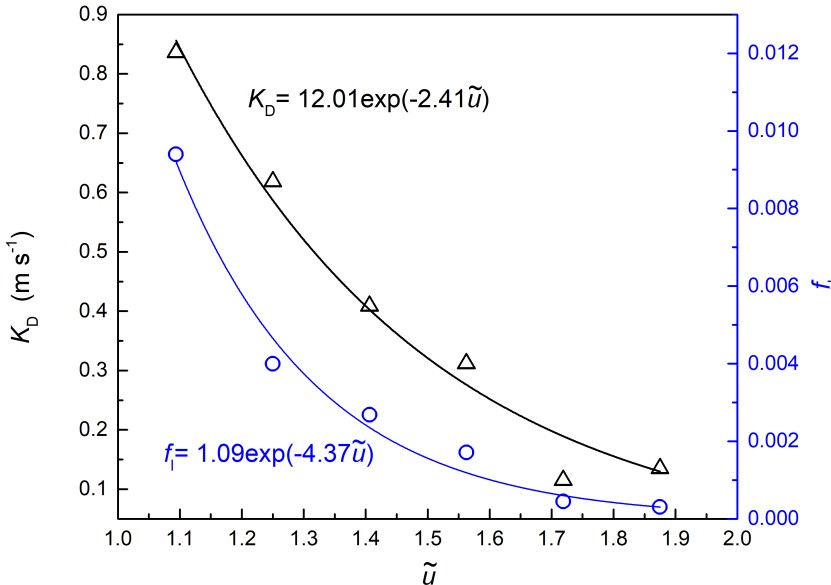

**Figure 8.** Deposition coefficient and the horizontal collection rate in all wind conditions. The solid lines are the fit curves.

## 4.2 Field predictions

To validate our conceptual model introduced in Section 4.1, we compare the results with two cases of field observations.

### 4.2.1 Case I: Comparison of suitable wind condition with Vogel et al. (2012)

Vogel et al. (2012, abbr. as VF2012) showed cornice evolution along the ridgeline of the Gruvefjellet plateau mountain above Nybyen in the period 2008–2010. They found that the cornice accretion happened during the entire snow seasons, when the averaged hourly maximum wind speeds exceeded 12 m s$^{-1}$, with a minimum of at least 10 m s$^{-1}$. It should be noted here that by analyzing the time series of wind speed data from Gruvefjellet meteo station (2022), the corresponding average wind speed is found to be $7.37 \pm 0.97$ m s$^{-1}$ when the maximum wind speed is in the range of 10.5 m s$^{-1}$ to 11.5 m s$^{-1}$. Thus, the friction velocity is 0.29 m s$^{-1}$ assuming a roughness length $z_0 = 10^{-4}$ m. This value is comparable to the threshold wind speed in previous research (Sugiura et al., 1998; JDoorschot et al., 2004; Clifton et al., 2006), considering the harder snow surface in Gruvefjellet (Eckerstorfer et al., 2013).

Considering the cornice accretion always appears in snowstorms, we assume that the snow transport rate $Q$ in the field can be expressed as the same value as its value in the saturated saltation (Vionnet et al., 2014):

$$Q = \frac{\rho_a}{g} u_*^3 (1 - \frac{u_{*t}^2}{u_*^2})(2.6 + 2\frac{u_{*t}}{u_*} + 2.5\frac{u_{*t}^2}{u_*^2}) \tag{12}$$

where $u_* = \frac{\kappa u_f}{ln(z_f/z_0)}$ is the friction velocity which is calculated with the field wind speed $u_f$ at height $z_f = 2.8$ m. $\kappa = 0.4$ is the Von Kármán constant, $g = 9.8$ m s$^{-2}$ is the gravitational acceleration and $z_0$ is the aerodynamic roughness length. $u_{*t}$ is the threshold friction velocity which is calculated based on the local threshold wind speed $u_t$.

Then we can estimate the potential maximum erosion rate as the aerodynamic entrainment rate by:

$$M_e = m_p \cdot \eta_{ae} \rho_a (u_*^2 - u_{*t}^2) \tag{13}$$

where $m_p = \frac{1}{6}\pi d_p^3 \rho_i$ is the mass of a snow particle, in which the average particle diameter $d_p$ in the field is assumed as 300 $\mu$m (Nishimura et al., 2014). $\eta_{ae} = 6 \times 10^5$ grains N$^{-1}$ s$^{-1}$ is an empirical parameter (Clifton and Lehning, 2008).

Considering the ratio of erosion rate in thickness and length $l_e/h_e$ is about 0.7, we can rewrite the length growth rate $l_g$ in Eq. (11) as:

$$l_g = \frac{Q f_1}{\rho_c d_p} - 0.7 M_e/\rho_c \tag{14}$$

Thus, we could infer that the length growth rate $l_g$ is only dependent on variables of the field wind speed $u_f$, the threshold wind speed $u_t$, and the roughness length $z_0$. To test the sensitivity of the input parameters, we choose different $z_0$ and $u_t$ to estimate the length growth rate in the wind speed range of VF2012, shown in Fig. 9. The automatic weather station in Gruvefjellet is located at $\sim$ 300 m from the cornice on the plateau. The wind station is at a flat field, and the roughness length $z_0$ can be assumed as the measurement values on the flat snow surface. The roughness lengths $z_0$ vary in snow covers (Clifton et al., 2006), typically over two orders of magnitude: from $10^{-5}$–$10^{-3}$ m for fresh snow in the field (Brock et al., 2006; König-Langlo, 1985). As is shown in Fig. 9, the roughness length and the threshold wind speed only have effects on the magnitude

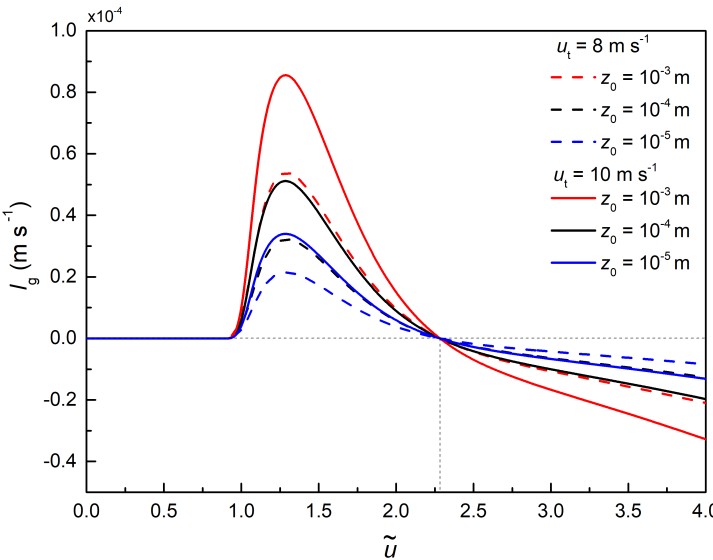

**Figure 9.** Estimations of cornice length growth rates in the fields.

of the maximum value of growth rates, while the suitable non-dimensional wind speed range remains the same. The predicted

wind range for snow cornice formation is about 1–2.3 times threshold wind speed, namely 10–22.6 m s$^{-1}$, which agrees with the field observations. And the maximum cornice growth is for winds about 30% higher than the threshold wind speed. There is no available length growth rate data in VF2012, so we use the following case to validate the length growth rate. For it is in the same site, we use $z_0 = 10^{-4}$ m and $u_t = 10$ m s$^{-1}$ in the following modeling.

### 4.2.2 Case II: Comparison of the length growth rate with Hancock et al. (2020)

Hancock et al. (2020, abbr. HF2020) used a Riegl® Laser Measurement Systems VZ-6000 ultra-long-range terrestrial laser scanner to repeatedly scan the Gruvefjellet and Platåberget cornice systems throughout the 2016–2017 and 2017–2018 winter seasons. Three cornice accretion events were recorded with the mean length growth rate over 10 mm h$^{-1}$, which is about 3.9–4.7×10$^{-6}$ m s$^{-1}$. However, in our experiment, the average length growth rate range is 1.2–2.7×10$^{-4}$ m s$^{-1}$. The main reason for the discrepancies between the laboratory and the field results is due to the temporally and spatially constrained estimations

of the threshold wind speeds for cornice accretion and cornice horizontal length growth rate (Vogel et al., 2012; Hancock et al., 2020). In the field, snow cornices have multiple growth periods in snowstorms that last a few hours. While in the RWT experiment, we mainly focused on a continuous growth process of a snow cornice. The fluctuating and intermittent wind in the field differs from the steady and stationary wind in the RWT, and this also causes the effective time for a cornice formation being much less than the sampling time (several hours to days). The sampling frequency is not sufficient to catch the complete

accretion period for the wind in the field is gusty and intermittent. Also, in the field, the cornice may partially collapse from time to time which is not recognized during the storm without any laser scanning. Thus, it is difficult to estimate the cornice length growth rate based on the daily averaged wind speed from HF2020. Here, we use the Weibull probability density function to reproduce a high-resolution time series of wind speed (Fig. 10a), which can be expressed as:

$$p(u) = (\frac{k}{\lambda})(\frac{u}{\lambda})^{k-1} e^{-(\frac{u}{\lambda})^k} \tag{15}$$

in which, $k$ is the shape factor which is normally between 1.5 to 3, depending on the wind variability. Smaller $k$ represents more gusty wind. For example, $k = 2$ represents for the moderately gusty wind (Seguro and Lambert, 2000). Here, we assumed it as 1.7. $\lambda$ is the scale factor which is calculated based on the daily averaged wind speed $\bar{u}$ and the gamma function of the inverse of the shape factor $k$:

$$\lambda = \frac{\bar{u}}{\Gamma(1 + \frac{1}{k})} \tag{16}$$

Figure 10a shows an example of wind speed time series produced by using Eq. 15 with a mean wind speed of 5.4 m s$^{-1}$ and time interval of 10 minutes. From the time series, we can estimate the length growth rate as follows:

$$l_g = \frac{1}{T} \int_0^T (\frac{Q(u) f_1(u)}{\rho_c \bar{d}_p} - 0.7 M_e(u)/\rho_c) dt \tag{17}$$

where $T$ is the sampling time of the scanner images. The transport rate $Q(u)$, horizontal collection rate $f_1(u)$, and the mass erosion rate $M_e(u)$ are the functions of wind speed $u$ in time series. To test the sensitivity of the time interval $dt$, we use values

of 5 min, 10 min, 15 min, 30 min, 1 hour, and 2 hours. The estimated growth rates are shown in Fig.10b. The length growth rate trends to a stable value when the time interval is shorter than 10 minutes. Thus, in the following analysis, we used 10 minutes as the time interval of wind data sampling.

Table. 3 shows the averaged length growth rates $l_g^f$ in three cornice accretion events in HF2020. The averaged length growth rates $l_g$ calculated from the model are comparable with that values from TLS data $l_g^f$ in the field (Hancock et al., 2020), which

indicates that our model has the potential ability to predict the cornice accretion in the field.

**Table 3.** Comparison results with field observations.

| Location | Dates | $l_g^f \times 10^{-6}$ (m s$^{-1}$) | $l_g \times 10^{-6}$ (m s$^{-1}$) |
|---|---|---|---|
| Plataberget | Feb17-Feb28 | 4.72 | 1.70±0.49 |
| Gruvefjellet | Jan12-Jan21 | 4.72 | 1.47±0.74 |
| Paltaberger | Apr25-May01 | 3.80 | 1.37±0.52 |

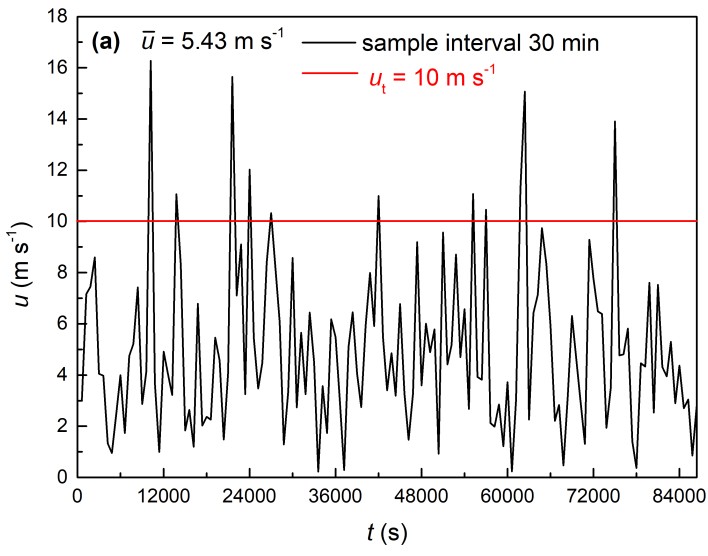

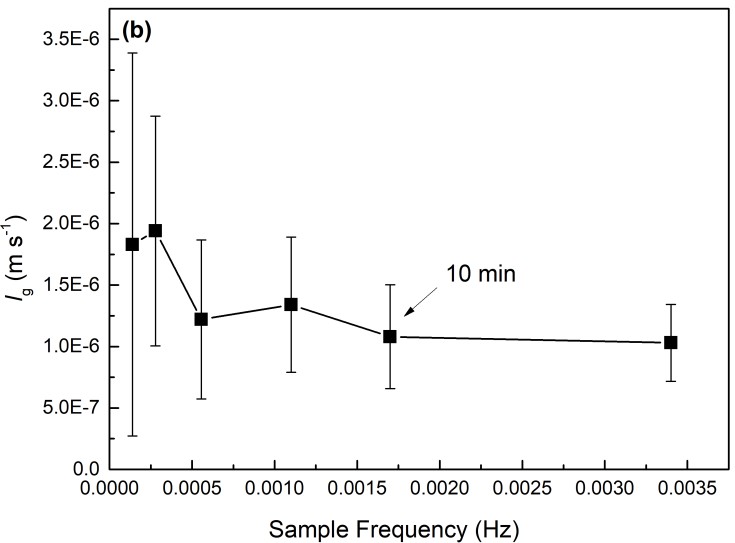

**Figure 10.** (a) Time series of wind speed $u$ in one day. (b) Average length growth rates in different sample frequencies.

What needs to be mentioned is that to enhance the model accuracy, the values of $K_d$ and $f_l$ inferred in this prediction model still need re-estimation and corrections for the natural larger-scale snow cornice. These two parameters may be influenced by the local topographical features. For future accurate field predictions on the cornice on a larger scale, more field measurement data are needed, such as the snowpack thickness on the root of a cornice, the mass concentration, the threshold friction velocity, roughness length, and cornice density to calculate the proper values for $K_d$ and $f_l$ in the field.

## 5   Conclusions and Outlook

We carried out RWT experiments and studied suitable wind conditions for cornice formation and growth. The results show that the snow cornices only grow at moderate wind speeds with a sufficient snow mass flux over the ridge of the model. The cornice growth process has two stages. The vertical growth rate of the cornice is typically lower relative to the horizontal growth. The mass collection efficiency decreases with the increasing wind speed and the corresponding drift rate, which can not be considered the indicator for cornice growth. Instead, the growth rates of cornice in length and thickness are determined by the combined effects of mass accumulation and erosion. The lower limit of wind speed is the threshold wind speed for snow transport, and the upper limit value of wind speed is when the deposition rate and the erosion rate arrive balanced. The most favorable wind condition for cornice growth is approximately 40% higher than the threshold wind speed for snow transport, at which the net deposition rate in length is maximum.

Based on the experimental results, a conceptual model is proposed for interpreting the mechanism of cornice growth. From this model, we can further explain that the magnitudes of the cornice thickness and length growth rates are determined by the competing effects of an increasing drift rate $Q$ and decreasing collection rates $K_d$ and $f_l$ with increasing wind speed above the threshold. Besides, based on the field observation data, such as roughness length, the threshold wind speed, and the local surface snow conditions, this model can be applied to field conditions to predict the cornice length growth rates and the suitable wind speed range. From the estimations at the study site of Gruvefjellet, we can conclude that the wind speed range of cornice growth is from 1–2.3 times the threshold wind speed, which is in line with the previous observations in the field. It is found that the most favorable wind condition for cornice growth is approximately 30% higher than the local threshold wind speed for this site. The discrepancies in the knowledge of the suitable wind speed range in the previous wind tunnel experiment and the field observations are mainly due to the differences in the local roughness lengths and the threshold wind speeds. In a future study, improvements of our model, such as predicting the snow cornice growth rates more accurately, still need higher frequency observation data on cornice growth and erosion and the measurements of other relevant parameters.

Overall, this study is a step forward in understanding the mechanism of cornice formation with detailed measurements and controlled environmental conditions. We also present progress in the methodology of observing snow cornice formation. In the future, this may lead to improvements in cornice-fall avalanche predictions.

*Author contributions.* YHX, LG, and BW designed the experiments. YHX and LG conducted the experiments, performed the data analysis, and prepared the first draft. ML, ZJ, and BW reviewed and edited the paper. HN and ML organized this study, contributed to its conceptualization, discussion, and finalized the paper.

*Competing interests.* The authors declare that they have no conflict of interest.

*Acknowledgements.* The authors would like to thank Matthias Jaggi for his Snow Maker expertise. The authors appreciate Dr. Mahdi Jafari, Daniela Brito Melo, Armin Sigmund for the valuable suggestions to improve the manuscript. The authors thank editor Melody Sandells, Holt Hancock, and the two anonymous referees for their constructive comments. This work was supported by the National Natural Science Foundation of China (grant no.: 42006187 and 41931179), the Second Tibetan Plateau Scientific Expedition and Research Program (grant no.: 2019QZKK020109-2), the Fundamental Research Funds for the Central Universities (grant no.: lzujbky-2021-it29). And the data and

code will be upload to Dryad repository after the paper is published.

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
