# Peer review of "Wind Conditions for Snow Cornice Formation in a Wind Tunnel"

_The Cryosphere, 2022_

## Referee Comment (RC2)

*General comments:*

I appreciate the opportunity to review the manuscript entitled "Environmental Conditions for Snow Cornice Formation tested in a Wind Tunnel." In this study, the authors seek to improve the process understanding of snow cornice formation by conducting wind tunnel experiments in a cold laboratory. Specifically, the authors simulate cornice development in the wind tunnel by forcing snow particles made by a snowmaker over a small "ridge" of compacted snow at various wind speeds. Cross-sectional photographs of the model ridge and associated cornices taken with high temporal resolution help illustrate cornice development under different wind speeds. The manuscript in its current form is generally well-written, with few grammatical errors and clear language. The authors describe their methodology such that future work can easily repeat, and thus build upon, the present experiments. Figures are relevant, clear, and appropriately described. I found the combination of the repeatable methodology and associated results to be a relevant basis for future field studies and would like to complement the authors on their work.

Relatively few, to my knowledge, studies in the last couple decades have addressed cornice formation in laboratory settings. I think such laboratory studies offer a compelling avenue to improve our understanding of cornice processes and refine conclusions derived from field data. The methods employed by the authors in the current study therefore have the potential to augment recent field investigations by better constraining the environmental conditions influencing various processes of cornice dynamics (e.g. wind speeds leading to cornice accretion). Such work falls within the scope of *The Cryosphere* and will be of interest to an audience of snow researchers and practitioners working with cornice-related avalanche problems.

In this context, although the manuscript provides a decent overview of some previous work and general cornice-related concepts, the current introduction does not, in my opinion, adequately address the scientific framework for the current study. Specifically, the introduction fails to effectively link the referenced field studies to the "macroscopic view" mentioned in the abstract and laboratory methods presented in the current work. The authors should, in my opinion, considerably expand the introduction to better introduce and justify the laboratory methods employed in this work as pathway to improve the understanding of cornice dynamics in the field. In such an expanded introduction, the authors would have an opportunity to cite the Naitou and Kobayashi paper referenced by reviewer #1 (which, to be fair, I also had not read previously) in addition to other laboratory experiments serving as a basis for the presented work. Additionally, the authors could help guide the reader by more specifically stating which aspects or processes of cornice formation they sought to investigate with their wind tunnel experiments – e.g. explicitly state what processes currently unresolved by field studies you hope to address in the laboratory. See also the specific comments related to content in the introduction.

My other major concern with the manuscript in its current form stems from the results and discussion in Section 3.2. In general, I think splitting the combined results and discussion section here could help with clarity (e.g. split the calculations and numerical results into a results section and the associated discussion into its own section). However, the main issue stems from the selection of the appropriate wind speed range for cornice growth in the field. The authors cite our 2020 paper as stating the wind speed range for cornice growth in the field is 12-30 m s$^{-1}$.

Unfortunately, this is a mischaracterization of the results from that paper. Vogel et al. (2012) determined cornice accretion occurred during periods with hourly maximum wind speeds of 12 m s$^{-1}$ and observed cornice scouring when maximum hourly wind speeds were as low as 15 m s$^{-1}$. In our 2020 work, the temporal constraints on our TLS measurements of cornice accretion were relatively poor and did not allow us to effectively determine a lower threshold wind speed for which cornice accretion begins to occur. Instead – and admittedly this is a weakness in that study – we simply used 5 m s$^{-1}$ as a conservative lower threshold for snow transport (and therefore, we assumed, cornice accretion) derived from the literature. Accordingly, although the comparison between the authors' experiments and field studies from lines 172 – 202 is interesting and relevant for this work, I think the authors should redo their calculations with a more appropriate $U_{t2.8}$ value.

My suggestion here would be to consider that field studies often struggle to determine the threshold wind speeds for cornice accretion and/or scouring due to temporal or spatial constraints on data acquisition. Laboratory experiments such as the current study can help better determine these thresholds, and therefore one option would be to extrapolate a "field" $U_{t2.8}$ based on your measured $U_{t0.4}$ and a logarithmic wind profile. This would then allow the authors to discuss how their results can help address gaps in field studies (e.g. more specifically constrain the wind speeds at which cornice accretion happens, with the wind speeds expressed for the height at which standard meteorological observations occur). In the conclusions, however, I would appreciate a link between your measurements (e.g. cornice growth occurs between 3.5 and 6 m s$^{-1}$) and the corresponding "field" wind speeds which will be more relevant, especially, for practitioners interested in your work.

***Specific comments:***

Title – I wonder if the title could be more specific than "environmental conditions" – would wind conditions be more appropriate?

Line 16 – please clarify what the percentages refer to, or consider omitting the percentages altogether

Lines 27-31 – consider splitting this long sentence for clarity and readability

Lines 36-38 – please revise this sentence, I don't understand what widely accepted hypothesis is referred to here

Lines 39-40 – which assumptions have no supporting evidence?

Lines 41-42 – this sentence needs to be revised in lieu of the existence of the Naitou and Kobayashi work. I am unable to read Japanese so cannot specifically comment on the methodological and content overlap between this work and the Naitou and Kobayashi study. I would suggest attempting to determine how your work differs from this previous work and adjusting the intro/results as needed.

Line 53 – is 7 m s$^{-1}$ the maximum wind speed this device can generate?

Lines 108-113 – Super cool! Thanks for this.

Figure 3 caption – I think the cornice length growth rate and cornice thickness growth rate are represented with Xs, not triangles.

Line 144 – by crackdown do you mean cornice collapse or failure?

Line 145-146 – Is this because higher wind speeds form a cornice with a smaller angle?

Lines 175-185 – I am struggling to follow these calculations here, which may partially be due to my inexperience with such work. Is it possible to more explicitly define your terms somewhere (e.g. in a table) in the manuscript to help along readers such as myself? Also, to what are you referencing the Leonard et al. (2012) study here?

Line 204 – please revise this sentence in lieu of Naitou and Kobayashi.

Lines 205 – 215 – specifically here I think this work would really benefit from explicitly linking your results to field meteorological measurements (e.g. $U_{t2.8}$) for increased utility of your results and work.

***Technical corrections:***

Line 28 – that snow cornices only grow

Line 135 – there are no more chances for slabs to form on the model edge because of…

Line 137 – gets smaller

Line 202 – the newly formed cornice

---

## Author Comment (AC1)

**Response to Referee #1's interactive comment**

1. Investigations of snow cornice development is worthwhile since its collapse is strongly related to the snow avalanche release; I cannot agree with you more. In this study, the leading-edge technology including the closed-circuit wind tunnel and the shadow graph imaging technologies. I appreciate very much for the efforts by authors.

**Response:** We thank the reviewer for a positive view on the importance of the subject covered by our submission.

2. However, that is all. Similar experiments in the wind tunnel were carried out more than 35 years ago by a master student as shown below and much more meaningful outcomes were obtained.

Naitou, A. and Kobayashi, D., Experimental Study on the Generation of a Snow Cornice, Low temperature science. Series A, Physical sciences, 44, 91-101, 1986.

https://eprints.lib.hokudai.ac.jp/dspace/bitstream/2115/18521/1/44_p91-101.pdf

Unfortunately, the text is written in Japanese. However, it cannot be an excuse, since English summary is attached, in which the wind speed of 4 to 8 m/s is suitable for the cornice formation, and the capture coefficient of drifting snow is also referred. Incidentally, I suppose some of the authors can recognize Chinese characters and are understandable what is mentioned in the test as well more or less. Please read through carefully.

**Response:** We are sorry that we didn't find this thesis before. Many thanks for mentioning this paper, and we have tried our best to translate and strive to understand the content correctly. In the revised manuscript we will highlight the innovations of our work and the progress compared with this paper.

3. Dependencies on not only the air temperature but also crystal shape, which are listed as the future work in the submitted manuscript, have been already studied. Thus, from my point of view, nothing looks new and no findings which deepen our understandings of the snow cornice formation mechanism are introduced in the submitted manuscript.

**Response:** We disagree that our paper does not present any new insights. No more relevant experimental work has been published except Naitou and Kobayashi's work as the reviewer mentioned. Considering this, we think there are still many open scientific questions on cornice formation. In Naitou and Kobayashi's paper, which has not gone through peer review, potential factors such as wind speed, air temperature, crystal shape, and mass flux have been investigated with respect to the phenomenon of cornice growth. However, the effects are still not fully understood with strong evidence and data from our point of view. In all, for cornice formation research, more detailed physical mechanism explanation and solid scientifical evidence are still lacking to prove the hypothesis.

A detailed comparison of five aspects, in which our work differs from Naitou and Kobayashi is listed below:

*1) The mechanism of cornice growth from the micro perspective.*

Naitou and Kobayashi believed that the cornice grew because, during the pause of the wind for ten seconds, particles stopped and attached to the cornice tip from the photo captured in the experiment. It is understandable that the intermittent flow

facilitates the formation of the cornice.

However, a new mechanism was exposed based on our experimental results, i.e., the cornice formation in a steady and continuous wind condition, as is shown in Figure 3(a) in the manuscript. From this perspective, cornice formation is a result of a dynamic mass balance between deposition and erosion. The particles stop and stick on the cornice because of sudden wind speed decreases near the edge.

*2) The suitable wind condition for cornice formation*

Both Naitou and Kobayashi's work and our experimental results proposed that snow cornice only grows under a suitable wind speed range. The values are not exactly the same which is caused by the effects of different snow surfaces and environmental conditions. The key point is that, compared with Naitou and Kobayashi's work, we proposed a clearer physical explanation for the suitable wind speed range and a reasonable method to link with field data.

In our study, we did the single variable (wind speed value) controlled experiments in the cold lab. We kept the air temperature as a constant value meanwhile changing the wind speed from 3 m s$^{-1}$ to 6.5 m s$^{-1}$ by the step of 0.5 m s$^{-1}$ continuously. We recorded the whole process of cornice formation using a camera and analyzed the data by an image analysis program automatically. We calculated the cornice growth rates both in length and thickness (while Naitou and Kobayashi only measured the bulk volume growth rate), as well as the erosion rates in length and thickness (after without seeding). Overall, we not only summarized the general rules for cornice growth but also explained the phenomenon that cornices grow at moderate wind speeds from the physical view of the interaction of deposition and erosion. Moreover, Naitou and Kobayashi proposed that the cornices grow only under 4-8 m s$^{-1}$ (for 1m height in the site field and 0.5 m in the wind tunnel), while Vogel (2012) proposed that the cornice can grow when the wind speed is between 12-30 m s$^{-1}$ and scouring can occur when maximum hourly wind speed is as low as 15 m s$^{-1}$. Hancock (2020) used 5 m s$^{-1}$ as a conservative lower threshold for cornice accretion, which is the threshold wind speed of snow particles entrainment. To figure out the huge gaps among these field observations, we proposed a non-dimensional method to link the gaps between the wind tunnel and the field observations, which could be used as the estimation in field on the threshold wind speed for snow cornice accretion or erosion.

From Naitou and Kobayashi's work, we can not generalize and use the wind at a certain height as an environmental parameter condition to determine the formation of snow cornice because the boundary layer flow state differs in different environments. Even with the simplest logarithmic profile to characterize the airflow near-surface, the differences in roughness lengths can lead to inconsistent conclusions.

Actually, the friction velocity is more closely related to the particle motion near the surface. Moreover, as a combined effect of the snow accumulation and erosion, cornice accretion is also associated with threshold friction velocity. Thus, the non-dimensional wind speed value proposed in our work is more universal.

*3) Experimental setup and instruments*

In our study, we recorded the whole cornice formation by implementing a state-of-art technology: the shadowgraphy method. We also created a particle-recognition

program that could analyze a series of images recorded automatically.

Compared to Naitou and Kobayashi's work, our instrument is more advanced, and the results allow for stronger interpretations. Using this method, we calculated the cornice growth rates in length and thickness, respectively, and found the link between the growth rate and the corresponding erosion/deposition rates.

*4) Effect of temperature and crystal shape on the cornice formation*

In Naitou and Kobayashi's work, they have shown the particle size distribution in different air temperature conditions using the shape factor method, and concluded that cornice grows under a temperature between 0 and -20 ℃, and the new snow in an irregular dendriform shape is more appropriate for the cornice formation than aged round snow.

We also tested the factors of wind speed, air temperature, and crystal shape that influence the cornice formation in the wind tunnel experiment. From the experimental results, we have concluded that the air temperature and snow crystal shape do have influences on the cornice formation process (which are not shown in this manuscript) but not as strong as the wind speed. Thus, we only analyzed the wind effects in the manuscript, and focuses on wind conditions and the internal physical mechanism of cornice formation. In the revised manuscript, we added these sentences: "Except for the suitable wind speed, snow cornice growth also depends on the air temperature and snow crystal shape which has been described by Naitou and Kobayashi (1986). To verify this point, we did the experiments with environmental temperatures of -20 ℃, -15 ℃, -10 ℃, -5 ℃, 0 ℃, and with snow of different densities and crystal shapes. The results showed that snow cornice can grow under all kinds of air temperatures, and fresh snow is more beneficial than old snow for cornice growth. However, due to the particle collision and fragmentation in the experiment during the experiment, it is difficult to precisely measure the crystal shape of each particle that will stick on the cornice. Thus, we do not in detail investigate crystal shape in this contribution."

*5) The collection rate of snow*

The collection rate is the ratio of cornice mass increase rate and the drift snow transport rate, which can reveal the efficiency of snow cornice formation with a particular value of mass flux of drift snow. In Naitou and Kobayashi's work, snow particles are collected in the pit below the cornice. The collection rate gets maximum when the $Q$ is between 1-10 g m$^{-1}$ s$^{-1}$, and its value is no more than 0.1. In our manuscript, the collection rate of cornice decreases with increasing drift rates for higher wind speeds. These two collection rates are actually the same in the method which is to calculate the captured ratio of particles that flow around the cornice. They are different in magnitude values because of different experimental system settings.

However, in our work, we proposed that the collection rate cannot determine the maximum growth rate of snow cornice. Actually, cornice growth is the combined effects of the deposition and erosion processes (shown in Figure 4(a) in the manuscript).

In summary, compared to Naitou and Kobayashi's work, we obtained the following new results, which deepens the understanding of the snow cornice

formation mechanism by quantitative analysis of the macro variables:

1) Offering an explanation why cornices only grow at suitable wind speeds. From the macro view, cornice growth is the dynamic mass balance between deposition and erosion. The wind speed range is limited by the comprehensive effects of drifting snow deposition and erosion on the edge of the ridge.

2) Finding different rates between cornice's horizontal and vertical growth rates in two stages of snow cornice growth.

3) Finding that the collection rate of snow cornice growth decreases with increasing wind speed, and it cannot determine the wind speed value for cornice maximum growth rate.

4) Proposing a comprehensive estimation method of the threshold wind speed value for the cornice accretion to link gaps between the field and wind tunnel experiment, combining the macro variables and the observation data.

4. Further, the discussions, in which authors argue the similarities between the wind tunnel experiments and the observations in the fields, look odd. As is common for the researchers working on blowing/drifting snow, the blowing threshold wind speed in nature is around 5 m/s at 2 to 3 m high (not 11 m/s!), which roughly corresponds to the friction velocity of 0.2 to 0.3 m/s. If you assume, Ut 0.4m=3.2 m/s in the wind tunnel corresponds to Ut2.8m=11m/s in the field, friction velocity and the roughness length will be calculated extremely large (roughly u*=2.4 m/s, z0=0.235m !!; in usual former should be 0.3 to 0.4 m/s and the latter the order of 10-4 m).  Thus, discussions below line 175 in this manuscript sound meaningless.

**Response:** Sorry for the misunderstanding we have caused. Indeed, in order to compare with the field site observation, the wind speeds were normalized using the threshold wind speed of cornice accretion in different cases. Actually, the measurement site of wind velocity is not directly over the ridge. For the field observation, it comes from an automatic weather station - Gruvefjellet meteorological station (464 m a.s.l.), located centrally on the plateau, 300 m from the study site, where the cornice study site is along the edge of the plateau mountain Gruvefjellet [~ 460 m above sea level (a.s.l.)]. For our experiment, the wind velocity is measured by MiniAir which is located at the entrance of S2, with a height of 0.4 m, while the cornice study area is 0.5 m downward, with a height of 0.125 m. From the experiment, we noticed that from the moment the drifting snow started, the cornice grew. Thus, we defined the threshold speed as the wind speed when drifting snow starts, as well as cornice starts to grow.

We used the **nondimensionalization method** to unify the wind speed range between field studies and our laboratory works, using the threshold speed. In our experiments, the cornice accretion starts after the wind speed exceeds the threshold wind speed ($3.2 \text{ m s}^{-1}$) with enough snow supply, and no cornice formation when the wind speed is below it (e.g., $3.0 \text{ m s}^{-1}$). In field observation, Vogel et al. (2012) surmised that cornice accretion proceeded during both entire snow seasons (46 h in 2008/2009, 54 h in 2009/2010), when wind speeds averaged $12 \text{ m s}^{-1}$, with a minimum of at least **$10 \text{ m s}^{-1}$**, marking the lower limit of the cornice accretion; Eckerstorfer et al. (2012) measured that the initial cornice accretion started along the plateau edge during the first snowfall (10-12, Oct. 2010) with

maximum wind speeds of **11 m s⁻¹**, which is lower than the observation result of Vogel et al. (2012). Thus, by analyzing the time series of wind speed data from Gruvefjellet meteo station (http://158.39.149.183/Gruvefjellet/index.html), we can conclude that the corresponding averaged wind speed is about $7.37 \pm 0.97$ m s⁻¹ (and mean friction velocity is about 0.288 m s⁻¹ using $z_0 = 10^{-4}$ m) when the maximum wind speed is in the range of 10.5~11.5 m s⁻¹. Considering the harder snow surface in Gruvefjellet (Eckerstorfer et al., 2012), this wind speed value is comparable to the threshold wind speed in previous literature. For the wind in the field is more gusty and turbulent compared to the wind tunnel, the actual threshold wind speed value for cornice growth in the non-dimensional calculation should equal the maximum value of the threshold wind speed measured in the field, according to the study of Li et al. (2020, DOI: 10.1029/2019GL086574). Thus, we chose the maximum wind speed value 11 m s⁻¹ as the comparison value with the threshold wind speed in our laboratory experiment where the wind condition is stable and steady.

The nondimensional velocity is then defined as: $\tilde{u} = \frac{u_{ref}}{u_{t\_ref}} = \frac{u_*}{u_{*t}}$, in which $u_{ref}, u_*$ are

the wind speed at the measurement height and friction velocity, respectively, and $u_{t\_ref}$, $u_{*t}$ are the corresponding threshold wind speed at the measurement height and threshold friction velocity, respectively.

To avoid misunderstanding, we revised the description of lines 172 to 178 as:
"Estimates of the threshold wind speeds for cornice accretion (Vogel et al., 2012; Hancock et al., 2020) are compromised by temporal or spatial constraints on data acquisition. Based on this, we made an estimation of the appropriate wind speed using a non-dimensional method to unify the wind speeds in the wind tunnel and fields. The non-dimensional mass concentration of snow particles can be estimated in the following steps:
1) Dimensionless wind speed $\tilde{u}$ can be calculated as the ratio of wind speed at the measured site relative to the threshold wind speed measured at the site when the

cornice starts to grow: $\tilde{u} = \frac{U_{0.4}}{U_{0.4t}} = \frac{U_{2.8}}{U_{2.8t}} = \frac{u_*}{u_{*t}}$. Here 0.4 m represents the sensor height in

our experiment, 2.8 m represents the weather station wind speed measurement setup height in Gruvefjellet, and * represents the friction velocity over the cornice. In the field observation, Vogel et al. (2012) surmised that cornice accretion proceeded during both entire snow seasons (46 h in 2008/2009, 54 h in 2009/2010), when wind speeds averaged 12 m s⁻¹, with a minimum of at least 10 m s⁻¹, marking the lower limit of the cornice accretion; Eckerstorfer et al. (2012) measured that the initial cornice accretion started along the plateau edge during the first snowfall (10-12, Oct. 2010) with maximum wind speeds of 11 m s⁻¹. Considering more gusty and turbulent winds in the field, the maximum value for threshold wind speed is more comparable to our laboratory data (wind flow is more stable), we chose 11 m s⁻¹ as the threshold wind speed in the field.

Thus, dimensionless snow transport rate on the flat surface $\widetilde{Q} = \frac{gQ}{\rho_a u_{*t}^3}$ can be calculated as

a function of the dimensionless wind velocity $\tilde{u}$. Several common formulas of the function are shown in Table 4."

We also added content to discussions:

"Our wind tunnel experiment results can resolve inconsistencies in these observations. From our wind tunnel experiment, we can conclude that the threshold wind speed for cornice accretion is very close to the threshold wind speed for particles entrained from the surface. The inconsistency in the threshold wind speed for cornice accretion is due to the different snow surface conditions. We can conclude from our experiment that: "Drifting snow is necessary for cornice formation. Only when the wind speed is over the threshold wind speed for particle entrainment, there exists an opportunity for particles to impact and stick on the edge surface, where accumulation is the basis for the cornice formation. When the non-dimensional wind speed $\tilde{u}$ is over 2.7, the scouring effect is much stronger than accretion so that no cornice forms, which is consistent with the Eckerstorfer et al. (2012) and Vogel et al. (2012) field observations."

5. Preferably, missing link between the 4 cm long and 5 mm thick cornice observed in the wind tunnel and the several-meter scale of cornice formed leeside of the mountain ridge should be also referred to answer the motivation in the introduction part.

**Response:** Thanks a lot for this suggestion. In the field, snow cornices form in snowstorms which can last a few hours and can have multiple growth periods during the snow season, which leads to a much larger scale. The shape and the size of the cornices are indeed an interesting topic, but our experiment here is not mainly focused on it. Due to the limitation of the field of view of the camera, our experiment didn't last until the final state of the snow cornice growth. In this work, we mainly focused on the laws of growth rates in the initial state and relevant physical explanations. We will make more proper instructions aiming at the research problems and the applicability of the results in the revised manuscript.

---

## Author Comment (AC2)

**Response to referee #2's interactive comment**

*General comments:*

1. I appreciate the opportunity to review the manuscript entitled "Environmental Conditions for Snow Cornice Formation tested in a Wind Tunnel." In this study, the authors seek to improve the process understanding of snow cornice formation by conducting wind tunnel experiments in a cold laboratory. Specifically, the authors simulate cornice development in the wind tunnel by forcing snow particles made by a snowmaker over a small "ridge" of compacted snow at various wind speeds. Cross-sectional photographs of the model ridge and associated cornices taken with high temporal resolution help illustrate cornice development under different wind speeds. The manuscript in its current form is generally well-written, with few grammatical errors and clear language. The authors describe their methodology such that future work can easily repeat, and thus build upon, the present experiments. Figures are relevant, clear, and appropriately described. I found the combination of the repeatable methodology and associated results to be a relevant basis for future field studies and would like to complement the authors on their work.

Relatively few, to my knowledge, studies in the last couple decades have addressed cornice formation in laboratory settings. I think such laboratory studies offer a compelling avenue to improve our understanding of cornice processes and refine conclusions derived from field data. The methods employed by the authors in the current study therefore have the potential to augment recent field investigations by better constraining the environmental conditions influencing various processes of cornice dynamics (e.g. wind speeds leading to cornice accretion). Such work falls within the scope of *The Cryosphere* and will be of interest to an audience of snow researchers and practitioners working with cornice-related avalanche problems.

**Response:** We thank the reviewer for the positive feedback on the general aspects of our work. We really appreciate your efforts in reviewing this manuscript. In the following, we respond point by point to your comments.

2. In this context, although the manuscript provides a decent overview of some previous work and general cornice-related concepts, the current introduction does not, in my opinion, adequately address the scientific framework for the current study. Specifically, the introduction fails to effectively link the referenced field studies to the "macroscopic view" mentioned in the abstract and laboratory methods presented in the current work. The authors should, in my opinion, considerably expand the introduction to better introduce and justify the laboratory methods employed in this work as pathway to improve the understanding of cornice dynamics in the field. In such an expanded introduction, the authors would have an opportunity to cite the Naitou and Kobayashi paper referenced by reviewer #1 (which, to be fair, I also had not read previously) in addition to other laboratory experiments serving as a basis for the presented work. Additionally, the authors could help guide the reader by more specifically stating which aspects or processes of cornice formation they sought to investigate with their wind

tunnel experiments – e.g., explicitly state what processes currently unresolved by field studies you hope to address in the laboratory. See also the specific comments related to content in the introduction.

**Response:** Thank you for this valuable suggestion. We have expanded the introduction by adding the following paragraph:

"Cornice accretion and scouring have only been measured in few field and laboratory environments: A notable exception is Naito and Kobayashi (1986), who proposed that the suitable wind speed for cornice growing process both in the wind tunnel and the field is between 4 to 8 m s$^{-1}$. In a moist Arctic environment, Vogel et al. (2012) determined that cornice accretion occurred during periods with hourly maximum wind speeds of 12 m s$^{-1}$ and observed cornice scouring when the maximum hourly wind speeds were as low as 15 m s$^{-1}$. Hancock et al. (2020) used 5 m s$^{-1}$ as a conservative lower threshold for snow transport for cornice accretion which is the threshold wind speed for particles for entrainment. To explain the huge gaps in the threshold value in field observation results and the inner growth mechanism from a physical point of view, wind tunnel experiments in a cold laboratory at WSL/SLF are presented to quantitatively investigate the cornice formation process."

In the discussion, we added:

"The gaps found between different observations are mainly caused by the different snow surface conditions. In Gruvefjellet, the snow surface is relatively hard, which means the threshold wind speed for snow particle entrainment is high. No matter what kind of snow surface there is, the threshold wind speed for cornice accretion can be roughly estimated by the threshold wind speed for particle entrainment. The threshold wind speed for cornice scouring can be roughly estimated at 2.75 times of threshold wind speed."

3. My other major concern with the manuscript in its current form stems from the results and discussion in Section 3.2. In general, I think splitting the combined results and discussion section here could help with clarity (e.g. split the calculations and numerical results into a results section and the associated discussion into its own section). However, the main issue stems from the selection of the appropriate wind speed range for cornice growth in the field. The authors cite our 2020 paper as stating the wind speed range for cornice growth in the field is 12-30 m s$^{-1}$.

Unfortunately, this is a mischaracterization of the results from that paper. Vogel et al. (2012) determined cornice accretion occurred during periods with hourly maximum wind speeds of 12 m s$^{-1}$ and observed cornice scouring when maximum hourly wind speeds were as low as 15 m s$^{-1}$. In our 2020 work, the temporal constraints on our TLS measurements of cornice accretion were relatively poor and did not allow us to effectively determine a lower threshold wind speed for which cornice accretion begins to occur. Instead – and admittedly this is a weakness in that study – we simply used 5 m s$^{-1}$ as a conservative lower threshold for snow transport (and therefore, we assumed, cornice accretion) derived from the literature. Accordingly, although the comparison between the authors' experiments and field studies from lines 172 – 202 is interesting and relevant for this work, I think the authors should redo their calculations with a more appropriate $U_{t2.8}$ value.

My suggestion here would be to consider that field studies often struggle to determine the threshold wind speeds for cornice accretion and/or scouring due to temporal or spatial constraints on data acquisition. Laboratory experiments such as the current study can help better determine these thresholds, and therefore one option would be to extrapolate a "field" $U_{t2.8}$ based on your measured $U_{t0.4}$ and a logarithmic wind profile. This would then allow the authors to discuss how their results can help address gaps in field studies (e.g. more specifically constrain the wind speeds at which cornice accretion happens, with the wind speeds expressed for the height at which standard meteorological observations occur). In the conclusions, however, I would appreciate a link between your measurements (e.g. cornice growth occurs between 3.5 and 6 m s⁻¹) and the corresponding "field" wind speeds which will be more relevant, especially, for practitioners interested in your work.

**Response:** Sorry for our misleading expressions using $U_{t0.4}$ and $U_{t2.8}$. Here, it is not suitable for us to use the logarithmic wind profile to link the laboratory experiments and field observations for the following reasons:

1) A well-developed boundary layer cannot be generated in the ring wind tunnel.

2) The wind speed sensor is installed at the height of 0.4 m in the inlet of section S2, 0.5 m upstream of the ridge edge, which is not the real wind speed over the cornice (0.125 m in height, as shown in Figure 1 ), while the field observation station is located 400 m upstream of the cornice study site that contains a 220 m plateau ridgeline, which is also not the real wind speed over the cornice, moreover there is a height difference between the field observation station (Gruvefjellet met station, 464 m a.s.l.) and the cornice study site (about 460 m a.s.l., Figure 2 in the paper of Vogel et al. 2012).

Thus, we used the **nondimensionalization method** to unify the wind speed range between field studies and our laboratory works, using the threshold speed. In our experiments, the cornice accretion starts after the wind speed exceeds the threshold wind speed (3.2 m s⁻¹) with enough snow supply, and no cornice formation when the wind speed is below it (e.g., 3.0 m s⁻¹). In field observation, Vogel et al. (2012) surmised that cornice accretion proceeded during both entire snow seasons (46h in 2008/2009, 54h in 2009/2010), when wind speeds averaged 12 m s⁻¹, with a minimum of at least **10 m s⁻¹**, marking the lower limit of the cornice accretion; Eckerstorfer et al. (2012) measured that the initial cornice accretion started along the plateau edge during the first snowfall (10-12, Oct. 2010) with maximum wind speeds of **11 m s⁻¹**, which is lower than the observation result of Vogel et al. (2012). Thus, by analyzing the time series of wind speed data from Gruvefjellet meteo station (http://158.39.149.183/Gruvefjellet/index.html), we can conclude that the corresponding averaged wind speed is about $7.37\pm 0.97$ m s⁻¹ (and mean friction velocity is about 0.288 m s⁻¹ using $z_0 = 10^{-4}$ m) when the maximum wind speed is in the range of 10.5~11.5 m s⁻¹. Considering the harder snow surface in Gruvefjellet (Eckerstorfer et al., 2012), this wind speed value is comparable to the threshold wind speed in previous literature. For the wind in the field is more gusty and turbulent compared to the wind tunnel, the actual threshold wind speed value for cornice growth in the non-dimensional calculation should equal the maximum value of the threshold wind speed measured in the field, according to the study of Li et al. (2020, DOI: 10.1029/2019GL086574). Thus, we chose the maximum wind speed value 11 m s⁻¹ as the

comparison value with the threshold wind speed in our laboratory experiment where the wind condition is stable and steady. The nondimensional velocity is then defined as: $\tilde{u} = \frac{u_{ref}}{u_{t\_ref}} = \frac{u_*}{u_{*t}}$, in which $u_{ref}$, $u_*$ are the wind speed at the measurement height and friction velocity, respectively, and $u_{t\_ref}$, $u_{*t}$ are the corresponding threshold wind speed at the measurement height and threshold friction velocity, respectively.

In summary, we used the **nondimensionalization method** to avoid all the problems of the uncertainty to choose the roughness length and reference height of the measured wind speed, and only need to figure out the threshold wind speed (when cornice accretion starts), which is a good way for extending our works to the field investigations.

To avoid misunderstanding, we revised the description of lines 172 to 178 as:

"Estimates of the threshold wind speeds for cornice accretion (Vogel et al., 2012; Hancock et al., 2020) are compromised by temporal or spatial constraints on data acquisition. Based on this, we made an estimation of the appropriate wind speed using a non-dimensional method to unify the wind speeds in the wind tunnel and fields. The non-dimensional mass concentration of snow particles can be estimated in the following steps:

1) Dimensionless wind speed $\tilde{u}$ can be calculated as the ratio of wind speed at the measured site relative to the threshold wind speed measured at the site when the cornice starts to grow: $\tilde{u} = \frac{U_{0.4}}{U_{0.4t}} = \frac{U_{2.8}}{U_{2.8t}} = \frac{u_*}{u_{*t}}$. Here 0.4 m represents the sensor height in our experiment, 2.8 m represents the weather station wind speed measurement setup height in Gruvefjellet, and * represents the friction velocity over the cornice. In the field observation, Vogel et al. (2012) surmised that cornice accretion proceeded during both entire snow seasons (46 h in 2008/2009, 54 h in 2009/2010), when wind speeds averaged 12 m s$^{-1}$, with a minimum of at least 10 m s$^{-1}$, marking the lower limit of the cornice accretion; Eckerstorfer et al. (2012) measured that the initial cornice accretion started along the plateau edge during the first snowfall (10-12, Oct. 2010) with maximum wind speeds of 11 m s$^{-1}$. Considering more gusty and turbulent winds in the field, the maximum value for threshold wind speed is more comparable to our laboratory data (wind flow is more stable), we chose 11 m s$^{-1}$ as the threshold wind speed in the field. Thus, dimensionless snow transport rate on the flat surface $\tilde{Q} = \frac{gQ}{\rho_a u_{*t}^3}$ can be calculated as a function of the dimensionless wind velocity $\tilde{u}$. Several common formulas of the function are shown in Table 4."

We also added content to discussions:

"Our wind tunnel experiment results can resolve inconsistencies in these observations. From our wind tunnel experiment, we can conclude that the threshold wind speed for cornice accretion is very close to the threshold wind speed for particles entrained from the surface. The inconsistency in the threshold wind speed for cornice accretion is due to the different snow surface conditions. We can conclude from our experiment that: "Drifting snow is necessary for cornice formation. Only when the wind speed is over the threshold wind speed for particle entrainment, there exists an opportunity for particles to impact and stick on the edge surface, where accumulation is

the basis for the cornice formation. When the non-dimensional wind speed $\tilde{u}$ is over 2.7, the scouring effect is much stronger than accretion so that no cornice forms, which is consistent with the Eckerstorfer et al. (2012) and Vogel et al. (2012) field observations."

*Specific comments:*

1. Title – I wonder if the title could be more specific than "environmental conditions" – would wind conditions be more appropriate?

**Response:** Thank you for the suggestion. We will revise the title to "Wind Conditions for Snow Cornice Formation tested in a Wind Tunnel."

2. Line 16 – please clarify what the percentages refer to, or consider omitting the percentages altogether

**Response:** The percentages refer to the contribution of different types that causes secondary snow avalanche in the snow seasons 2006-2009 in Longyearbyen. For clarity, we have omitted the percentages in the revised manuscript.

3. Lines 27-31 – consider splitting this long sentence for clarity and readability

**Response:** Thank you for pointing this out, we will revise it as:

"Indirect evidence was presented by van Herwijnen and Fierz (2014) that snow cornice only grows under moderate to high strength wind during or soon after the snowfall. The cornice width from observation is in remarkable agreement with the wind drift index calculated by the snow cover model SNOWPACK (Lehning and Fierz, 2008), which indicates that snow transport plays an important role in cornice formation."

4. Lines 36-38 – please revise this sentence, I don't understand what widely accepted hypothesis is referred to here

**Response:** We have deleted this sentence which might lead to misunderstanding: "Particles, which follow the changing wind direction locally as the flow passes the ridge will stick to the growing cornice front at the mountain ridge is a widely accepted hypothesis."

5. Lines 39-40 – which assumptions have no supporting evidence?

**Response:** In here, we mean these assumptions of the vortex and the electric fields lack supporting experimental evidence. To make it clearer, we revised the sentence to "However, there is no direct experimental evidence to support neither vortex nor electric field influences on the cornice formation."

6. Lines 41-42 – this sentence needs to be revised in lieu of the existence of the Naitou and Kobayashi work. I am unable to read Japanese so cannot specifically comment on the methodological and content overlap between this work and the Naitou and Kobayashi study. I would suggest attempting to determine how your work differs from this previous work and adjusting the intro/results as needed.

**Response:** Thank you very much for the suggestion. We have answered the differences between our work and Naitou and Kobayashi's study in the reply to the first reviewer (*Q3*) and will adjust the intro/results in the revised manuscript according to the context of the discussion of the reply.

7. Line 53 – is 7 m s$^{-1}$ the maximum wind speed this device can generate?

**Response:** The maximum wind speed can reach 8 m s$^{-1}$. However, in our experiment, we only use the range of 3 - 7 m s$^{-1}$ for there is no cornice when $U \leq 3$ m s$^{-1}$ and $U \geq$

6.5 m s$^{-1}$.

8. Lines 108-113 – Super cool! Thanks for this.

**Response:** Thanks for your field observation data.

9. Figure 3 caption – I think the cornice length growth rate and cornice thickness growth rate are represented with Xs, not triangles.

**Response:** Thank you for pointing this out, we will correct the caption to "cornice length growth rate (light blue Xs), cornice thickness growth rate (grey Xs)".

10. Line 144 – by crackdown do you mean cornice collapse or failure?

**Response:** Thank you for pointing this out, we will change the word "crackdown" to "collapse" for precise expression in the revision.

11. Line 145-146 – Is this because higher wind speeds form a cornice with a smaller angle?

**Response:** Yes, cornices form at a higher wind speed with a smaller angle.

12. Lines 175-185 – I am struggling to follow these calculations here, which may partially be due to my inexperience with such work. Is it possible to more explicitly define your terms somewhere (e.g. in a table)s in the manuscript to help along readers such as myself? Also, to what are you referencing the Leonard et al. (2012) study here?

**Response:** Thanks for your comment, we will add a notation with physical definitions and units of all the symbols in the manuscript.
For Leonard et al. (2012) study, we were referencing the threshold friction velocity. However, we deleted the sentence "where $u_{*t} = 0.25$ m s$^{-1}$ is the threshold friction velocity Leonard et al. (2012)" to avoid misunderstanding.

13. Line 204 – please revise this sentence in lieu of Naitou and Kobayashi.

**Response:** We have deleted the word "first" in this sentence.

14. Lines 205 – 215 – specifically here I think this work would really benefit from explicitly linking your results to field meteorological measurements (e.g. Ut2.8) for increased utility of your results and work.

**Response:** Thanks for this valuable suggestion. We will reorganize it by splitting the linking wind tunnel results to field meteorological measurements as a separate section of the Discussion.

15. Technical corrections:
Line 28 – that snow cornices only grow
Line 135 – there are no more chances for slabs to form on the model edge because of…
Line 137 – gets smaller
Line 202 – the newly formed cornice

**Response:** Thank you for pointing these out. We will revise them in the revised manuscript.

---

## Author Response (AR1)

Dear Editor:

Thank you for your kind letter of "TC-2022-27 (author) - file upload for peer-review completion" on May 25, 2022. We have revised the manuscript "Wind Conditions for Snow Cornice Formation in a Wind Tunnel" in accordance with the reviewers' comments and carefully proofread the manuscript to minimize grammatical and bibliographical errors during the last two months. Below is our revision description according to the reviewers' comments. The font color of the comments is black, the responses in blue, and the revisions in red.

**Point-By-Point Reply to Editor's Comments**

1. Thank you for your extensive and detailed response to the reviewer comments, which is really helpful. The main issue is the dissertation on this topic, which you indicate is sufficiently different from your study for a variety of reasons. Please could you go ahead and amend your paper as you suggest with extra attention paid to the comparison between this study and the dissertation.

**Response**: We are grateful to the editor for her encouraging comments and constructive suggestions. We appreciate she gave us the opportunity to resubmit the revised manuscript and the chance to make a comparison between our work and the Japanese dissertation of Naito and Kobayashi (1986) (abbreviated as N&K86 below). Here we highlight our differences:

(1) So far, there are few laboratory experiments on cornice formation, except for the work of N&K86. However, due to the language gap, few researchers have noticed this non-peer-reviewed Japanese dissertation. Our work can not only enhance the reliability of N&K86 by reproducing their results but also attract more researchers' attention to the snow cornice topic through full peer review and open discussions, which is still valuable.

(2) We recorded the whole cornice formation by implementing a state-of-art technology: the shadowgraphy method, which allowed us to present more details on snow cornice formation. For example, we quantitively measured the whole growth process of snow cornice. We recorded the time series of cornice growth rates in length and thickness and particle mass flux in the air, etc. These experimental data are fundamental for further conceptual investigation of cornice formation.

(3) We proposed a conceptual model to explain the length and thickness growth of snow cornice based on the mass balance conservation and the experimental results, which have a good consistent with the field observation results in Gruvefjellet, Svalbard, Norway (Vogel et al., 2012; Hancock et al., 2020).

(4) We also tested the factors of the air temperature and grain shape on the snow cornice formation. However, these effects are not dominant as the wind speed. Thus, in this manuscript, we mainly focused on the wind effects and didn't mention these results.

More details can be seen in the revised manuscript. For example, we summarized their work and pointed out the open scientific questions in lines 30-31: "Naito and Kobayashi (1986) measured the suitable wind speed for cornice formation is between

4 m s$^{-1}$ to 8 m s$^{-1}$, at 1 m above the snow surface in the field and at the center (0.5 m height) in the wind tunnel." and lines 46-53: "There are few laboratory experiments on cornice formation except Naruse (1985) and Naito and Kobayashi (1986). Naito and Kobayashi (1986) carried out experiments both in the wind tunnel and in the field, observing the process of snow cornice. They described the snow cornice formation as a process that snow particles adhere one after another at the leeward edge, in the form of a thin slab of snow elongating leeward, then the slab hangs down under its weight, depositing drifted snow particles on it. However, quantitative descriptions of this process have not been reported. Their results show that the cornice growth under suitable conditions of the air temperature is between -20 ºC to 0 ºC, the wind speed is between 4 m s$^{-1}$ to 8 m s$^{-1}$, and fresh snow with an irregular dendritic shape. However, further quantitative analysis of experiments has not been carried out."

In section 3.2, we added the comparison results of our collection rates value in lines 165-169:

"A non-dimensional wind speed $\tilde{u} = \frac{u}{u_t}$ is defined here to compare with the experimental results of Naito and Kobayashi (1986). In this definition, $u_t$ is the threshold wind speed which can be considered as the lower limit wind speed value for cornice growth. As is shown in Fig. 5, the mass collection efficiency in both experiments decreases with the increasing wind speed and the corresponding drift rate. Our experimental results are much larger than that in N&K86, which is mainly due to the different wind tunnel sizes."

[Figure]

Fig. 5 Collection efficiency $E$ (in blue) and snow transport rate $Q$ (in red) under different non-dimensional wind speeds $\tilde{u}$. X represents the distance from the snow particle feeding point to the mass collection pits where the cornice grows. Lines are for ring

wind-tunnel experiments, hollow scatters are for N&K86. N&K86 represents the experiment results of (Naito and Kobayashi, 1986).

2. I recommend one further change to remove the word 'testing' from your new proposed title so that it reads 'Wind Conditions for Snow Cornice Formation in a Wind Tunnel'.
**Response**: Thanks for your insightful suggestion. We have revised the title to:
"Wind Conditions for Snow Cornice Formation in a Wind Tunnel".

**Point-By-Point Reply to Referee #1's Comments**

1. Investigations of snow cornice development is worthwhile since its collapse is strongly related to the snow avalanche release; I cannot agree with you more. In this study, the leading-edge technology including the closed-circuit wind tunnel and the shadow graph imaging technologies. I appreciate very much for the efforts by authors.

**Response:** We thank the reviewer for a positive view on the importance of the subject covered by our submission.

2. However, that is all. Similar experiments in the wind tunnel were carried out more than 35 years ago by a master student as shown below and much more meaningful outcomes were obtained.

Naitou, A. and Kobayashi, D., Experimental Study on the Generation of a Snow Cornice, Low temperature science. Series A, Physical sciences, 44, 91-101, 1986.

https://eprints.lib.hokudai.ac.jp/dspace/bitstream/2115/18521/1/44_p91-101.pdf

Unfortunately, the text is written in Japanese. However, it cannot be an excuse, since English summary is attached, in which the wind speed of 4 to 8 m/s is suitable for the cornice formation, and the capture coefficient of drifting snow is also referred. Incidentally, I suppose some of the authors can recognize Chinese characters and are understandable what is mentioned in the test as well more or less. Please read through carefully.

**Response:** We are sorry that we didn't find this thesis before. Many thanks for mentioning this paper, and we have tried our best to translate and strive to understand the content correctly. In the revised manuscript, we highlighted the innovations of our work and the progress compared to this paper. We have introduced this work in the introduction section, in lines 46-53: "There are few laboratory experiments on cornice formation except Naruse (1985) and Naito and Kobayashi (1986). Naito and Kobayashi (1986) carried out experiments both in the wind tunnel and in the field, observing the process of snow cornice. They described the snow cornice formation as a process that snow particles adhere one after another at the leeward edge, in the form of a thin slab of snow elongating leeward, then the slab hangs down under its weight, depositing drifted snow particles on it. However, quantitative descriptions of this process have not been reported. Their results show that the cornice growth under suitable conditions of the air temperature is between -20 ºC to 0 ºC, the wind speed is between 4 m s$^{-1}$ to 8 m s$^{-1}$, and fresh snow with an irregular dendritic shape. However, further quantitative analysis of experiments has not been carried out."

And in section 3.2, we added the comparison results of our collection rates value in lines 165-169: "A non-dimensional wind speed $\tilde{u} = \frac{u}{u_t}$ is defined here to compare with the experimental results of Naito and Kobayashi (1986). In this definition, $u_t$ is the threshold wind speed which can be considered as the lower limit wind speed value for cornice growth. As is shown in Fig. 5, the mass collection efficiency in both experiments decreases with the increasing wind speed and the corresponding drift rate. Our experimental results are much larger than that in N&K86, which is mainly due to the different wind tunnel sizes."

[Figure]

Fig. 5 Collection efficiency $E$ (in blue) and snow transport rate $Q$ (in red) under different non-dimensional wind speeds $\tilde{u}$. X represents the distance from the snow particle feeding point to the mass collection pits where the cornice grows. Lines are for ring wind-tunnel experiments, hollow scatters are for N&K86. N&K86 represents the experiment results of (Naito and Kobayashi, 1986).

3. Dependencies on not only the air temperature but also crystal shape, which are listed as the future work in the submitted manuscript, have been already studied. Thus, from my point of view, nothing looks new and no findings which deepen our understandings of the snow cornice formation mechanism are introduced in the submitted manuscript. **Response:** We disagree that our paper does not present any new insights. No more relevant experimental work has been published except N&K86's work as the reviewer mentioned. Considering this, we think there are still many open scientific questions on cornice formation. In N&K86's paper, which has not gone through peer review, potential factors such as wind speed, air temperature, crystal shape, and mass flux have been investigated with respect to the phenomenon of cornice growth. However, the effects are still not fully understood with solid evidence and data from our point of view. In all, for cornice formation research, more detailed physical mechanism explanations and solid scientifical evidence are still lacking. A detailed comparison of four aspects is presented in reply to Editor.

In summary, compared to N&K86's work, we obtained the following new results, which deepens the understanding of the snow cornice formation mechanism by quantitative analysis of the macro variables:
1) Offering an explanation of the cornices only growing at suitable wind speeds. From the macro view, cornice growth is the dynamic mass balance between deposition and erosion. The wind speed range is limited by the comprehensive effects of drifting snow

deposition and erosion on the edge of the ridge. The lower limit wind speed for cornice growth is approximately equal to the threshold wind speed for transport. The upper limit of wind speed is when the erosion rate is over the pure deposition rate.

2) Finding the cornice growth process has two stages: In the first stage, a thin slab grows and overhangs at the edge. In the second stage, cornice thickness and length both increase simultaneously.

3) The cornice only grows at a moderate wind speed range (1-2.03 $\tilde{u}$). The length growth rate gets maximum at the wind speed is 40 % over the threshold.

4) Finding that the collection rates of snow cornice growth decrease with increasing wind speed, and it cannot directly reflect the cornice growth characteristics. Instead, the pure deposition rates, the erosion rates, and the growth rates both in length and thickness were analyzed separately for all wind conditions. From the results, we can conclude that in all wind conditions, the cornice starts to grow when the wind speed exceeds the threshold value and starts to scour when the erosion rate is over the pure deposition rate.

5) Setting up a conceptual model to estimate the suitable wind speed range for cornice growth and the length growth rate of a cornice. The estimation results are in good consistency with the field observation results.

4. Further, the discussions, in which authors argue the similarities between the wind tunnel experiments and the observations in the fields, look odd. As is common for the researchers working on blowing/drifting snow, the blowing threshold wind speed in nature is around 5 m/s at 2 to 3 m high (not 11 m/s!), which roughly corresponds to the friction velocity of 0.2 to 0.3 m/s. If you assume, Ut 0.4m=3.2 m/s in the wind tunnel corresponds to Ut2.8m=11m/s in the field, friction velocity and the roughness length will be calculated extremely large (roughly u*=2.4 m/s, z0=0.235m !!; in usual former should be 0.3 to 0.4 m/s and the latter the order of 10-4 m).  Thus, discussions below line 175 in this manuscript sound meaningless.

**Response:** Sorry for the misunderstanding we have caused. 0.4 m and 2.8 m are the heights of the wind speed sensors. However, both in our experiment and in the field the sensors are not directly over the ridge. For our experiment, the wind velocity is measured by a MiniAir sensor installed at the entrance of S2, while the cornice study area is 0.5 m downward, with a height of 0.125 m. For the field observation, the wind speed data comes from an automatic weather station - Gruvefjellet meteorological station (464 m a.s.l.), located centrally on the plateau, 300 m from the study site, where the cornice study site is along the edge of the plateau mountain Gruvefjellet (~460 m a.s.l.). Thus, the logarithmic wind profiles are not directly available in this situation. Meanwhile, we observed that the cornice grew since the drifting snow started our experiment. Thus, we defined the threshold wind speed $u_t$ as the wind speed when drifting snow starts, namely the wind speed that a cornice starts to grow.

We used the **nondimensionalization method** to unify the wind speed range between field studies and our laboratory works by using $u_t$. In our experiments, the cornice accretion starts after the wind speed exceeds 3.2 m s$^{-1}$ with enough snow supply, and no cornice formation when the wind speed is below it (e.g., 3.0 m s$^{-1}$). In field observation, Vogel et al. (2012) surmised that cornice accretion proceeded during both

entire snow seasons (46 h in 2008/2009, 54 h in 2009/2010), when hourly maximum wind speed is averaged of 12 m s$^{-1}$, with a minimum of at least **10 m s$^{-1}$**, marking the lower limit of the cornice accretion. Eckerstorfer et al. (2012) measured that the initial cornice accretion started along the plateau edge during the first snowfall (10-12, Oct. 2010) with a maximum wind speed of **11 m s$^{-1}$**. We found that the corresponding averaged wind speed is about 7.37± 0.97 m s$^{-1}$ when the maximum wind speed is in the range of 10.5~11.5 m s$^{-1}$, by analyzing the time series of wind speed data from Gruvefjellet meteo station (http://158.39.149.183/Gruvefjellet/index.html), and the friction velocity is about 0.288 m s$^{-1}$ by using $z_0 = 10^{-4}$ m. Considering the harder snow surface in Gruvefjellet (Eckerstorfer et al., 2012), this wind speed value is comparable to the threshold wind speed in previous literature. Considering the wind in the field is gustier and more turbulent compared to the wind tunnel, the actual threshold wind speed value for cornice growth in the non-dimensional calculation should equal the maximum value of the threshold wind speed measured in the field, according to the study of Li et al. (2020, DOI: 10.1029/2019GL086574). Thus, we chose the maximum wind speed value of 10 m s$^{-1}$ as the threshold wind velocity.

To avoid misunderstanding, we set up a new conceptual model for interpreting the inner mechanism of cornice growth and compared the estimation cornice growth rate with the field observation in section 4 Discussion.

5. Preferably, missing link between the 4 cm long and 5 mm thick cornice observed in the wind tunnel and the several-meter scale of cornice formed leeside of the mountain ridge should be also referred to answer the motivation in the introduction part.

**Response:** Thanks for this insightful suggestion. In the field, snow cornices form in snowstorms that can last a few hours and can have multiple growth periods during the snow season, which leads to a much larger scale. The shape and the size of the cornices are indeed an interesting topic, but our experiment here is not mainly focused on it. Due to the limitation of the field of view of the camera, our experiment didn't last until the final state of the snow cornice growth. In this work, we mainly focused on the laws of growth rates in the initial state and relevant physical explanations.

In the revised manuscript, we added the link in the introduction section from lines 58-60:

"Therefore, wind tunnel experiments with controlled environmental conditions and quantitative descriptions of the individual cornice formation process as a pathway to improve the understanding of cornice dynamics in the field, particularly on the wind effects on cornice formation, are essential."

And in the results section from lines 202-204:

"The presented framework for characterizing cornice accretion may provide a basis for future field and laboratory studies under different conditions."

Moreover, based on the experimental results, we proposed a conceptual model (in the discussion section) that can explain the mechanism for cornice growth in length and thickness. We estimated the suitable wind speed range and the growth rate of the cornice in the field and compared the results with the field observations from Vogel et al. (2012) and Hancock et al. (2020), which are in the good consistent.

**Point-By-Point Reply to Referee #2's Comments**

*General comments:*

1. I appreciate the opportunity to review the manuscript entitled "Environmental Conditions for Snow Cornice Formation tested in a Wind Tunnel." In this study, the authors seek to improve the process understanding of snow cornice formation by conducting wind tunnel experiments in a cold laboratory. Specifically, the authors simulate cornice development in the wind tunnel by forcing snow particles made by a snowmaker over a small "ridge" of compacted snow at various wind speeds. Cross-sectional photographs of the model ridge and associated cornices taken with high temporal resolution help illustrate cornice development under different wind speeds. The manuscript in its current form is generally well-written, with few grammatical errors and clear language. The authors describe their methodology such that future work can easily repeat, and thus build upon, the present experiments. Figures are relevant, clear, and appropriately described. I found the combination of the repeatable methodology and associated results to be a relevant basis for future field studies and would like to complement the authors on their work.

Relatively few, to my knowledge, studies in the last couple decades have addressed cornice formation in laboratory settings. I think such laboratory studies offer a compelling avenue to improve our understanding of cornice processes and refine conclusions derived from field data. The methods employed by the authors in the current study therefore have the potential to augment recent field investigations by better constraining the environmental conditions influencing various processes of cornice dynamics (e.g. wind speeds leading to cornice accretion). Such work falls within the scope of *The Cryosphere* and will be of interest to an audience of snow researchers and practitioners working with cornice-related avalanche problems.

**Response:** We thank the reviewer for the positive feedback on the general aspects of our work. We really appreciate your efforts in reviewing this manuscript. In the following, we respond point by point to your comments.

2. In this context, although the manuscript provides a decent overview of some previous work and general cornice-related concepts, the current introduction does not, in my opinion, adequately address the scientific framework for the current study. Specifically, the introduction fails to effectively link the referenced field studies to the "macroscopic view" mentioned in the abstract and laboratory methods presented in the current work. The authors should, in my opinion, considerably expand the introduction to better introduce and justify the laboratory methods employed in this work as pathway to improve the understanding of cornice dynamics in the field.

**Response**: Thanks for your insightful suggestion. We linked the fieldwork with the experimental research by pointing out the constraints from the field observation and wind tunnel experiment as the pathway to improving our understanding of the cornice dynamics. In the revised manuscript, we added the link in the introduction section from lines 58-60:

"Therefore, wind tunnel experiments with controlled environmental conditions and

quantitative descriptions of the individual cornice formation process as a pathway to improve the understanding of cornice dynamics in the field, particularly on the wind effects on cornice formation, are essential."

And in the results section from lines 201-203:

"The presented framework for characterizing cornice accretion may provide a basis for future field and laboratory studies under different conditions."

3. In such an expanded introduction, the authors would have an opportunity to cite the Naitou and Kobayashi paper referenced by reviewer #1 (which, to be fair, I also had not read previously) in addition to other laboratory experiments serving as a basis for the presented work.

   **Response**: Thanks for your advice. We have introduced the work of Naito and Kobayashi in the revised manuscript, in lines 46-53: "There are few laboratory experiments on cornice formation except Naruse (1985) and Naito and Kobayashi (1986). Naito and Kobayashi (1986) carried out experiments both in the wind tunnel and in the field, observing the process of snow cornice. They described the snow cornice formation as a process that snow particles adhere one after another at the leeward edge, in the form of a thin slab of snow elongating leeward, then the slab hangs down under its weight, depositing drifted snow particles on it. However, quantitative descriptions of this process have not been reported. Their results show that the cornice growth under suitable conditions of the air temperature is between -20 $^{\circ}$C to 0 $^{\circ}$C, the wind speed is between 4 m s$^{-1}$ to 8 m s$^{-1}$, and fresh snow with an irregular dendritic shape. However, further quantitative analysis of experiments has not been carried out."

4. Additionally, the authors could help guide the reader by more specifically stating which aspects or processes of cornice formation they sought to investigate with their wind tunnel experiments – e.g., explicitly state what processes currently unresolved by field studies you hope to address in the laboratory. See also the specific comments related to content in the introduction.

**Response:** Thank you for this valuable suggestion. We have rewritten the section introduction by directly pointing out these open questions currently unresolved by the field studies, and we investigated them in the RWT experiments. In the introduction, we pointed out the open questions in lines 36-37: "However, to our best knowledge, this discrepancy and the conditions under which certain wind speed ranges apply have not been investigated." lines 43-46: "Due to the compromise of these field observations, continuous observations on individual cornice accretion and failure events are hard to achieve (Hancock et al., 2020). Specifically, measuring the horizontal growth of snow cornice (Vogel et al., 2012) and recording dynamic details of snow mass transport simultaneously is hard to achieve." lines 52-56: "However, further quantitative analysis of experiments has not been carried out. Mott et al. (2010) have indicated that snow cornice formation is mainly through snow distribution processes driven by saltation. However, due to the lack of physical mechanism of snow cornice formation, cornice characteristic features could not be reproduced in numerical simulation of snow distribution in mountain areas (Gauer2001). Thus, there is still no snow cornice prediction model that could be used in avalanche prevention so far."

5. My other major concern with the manuscript in its current form stems from the results

and discussion in Section 3.2. In general, I think splitting the combined results and discussion section here could help with clarity (e.g. split the calculations and numerical results into a results section and the associated discussion into its own section).

**Response:** Thanks for your suggestion. We have split the result section into two parts. Now the result section is only for the presentation of the experimental data of cornice formation, such as length growth rate. The discussion section proposed a conceptual model based on the experimental results, as well as its application in the field. Please see more details in the revised manuscript.

6. However, the main issue stems from the selection of the appropriate wind speed range for cornice growth in the field. The authors cite our 2020 paper as stating the wind speed range for cornice growth in the field is 12-30 m s$^{-1}$.

Unfortunately, this is a mischaracterization of the results from that paper. Vogel et al. (2012) determined cornice accretion occurred during periods with hourly maximum wind speeds of 12 m s$^{-1}$ and observed cornice scouring when maximum hourly wind speeds were as low as 15 m s$^{-1}$. In our 2020 work, the temporal constraints on our TLS measurements of cornice accretion were relatively poor and did not allow us to effectively determine a lower threshold wind speed for which cornice accretion begins to occur. Instead – and admittedly this is a weakness in that study – we simply used 5 m s$^{-1}$ as a conservative lower threshold for snow transport (and therefore, we assumed, cornice accretion) derived from the literature. Accordingly, although the comparison between the authors' experiments and field studies from lines 172 – 202 is interesting and relevant for this work, I think the authors should redo their calculations with a more appropriate $U_{t2.8}$ value.

My suggestion here would be to consider that field studies often struggle to determine the threshold wind speeds for cornice accretion and/or scouring due to temporal or spatial constraints on data acquisition. Laboratory experiments such as the current study can help better determine these thresholds, and therefore one option would be to extrapolate a "field" $U_{t2.8}$ based on your measured $U_{t0.4}$ and a logarithmic wind profile. This would then allow the authors to discuss how their results can help address gaps in field studies (e.g. more specifically constrain the wind speeds at which cornice accretion happens, with the wind speeds expressed for the height at which standard meteorological observations occur). In the conclusions, however, I would appreciate a link between your measurements (e.g. cornice growth occurs between 3.5 and 6 m s$^{-1}$) and the corresponding "field" wind speeds which will be more relevant, especially, for practitioners interested in your work.

**Response:** Sorry for our misleading expressions using $U_{t0.4}$ and $U_{t2.8}$. Here, it is not suitable for us to use the logarithmic wind profile to link the laboratory experiments and field observations for the following reasons:

   1) A well-developed boundary layer cannot be generated in the ring wind tunnel.

   2) The wind speed sensor is installed at the height of 0.4 m in the inlet of section S2, 0.5 m upstream of the ridge edge, which is not the real wind speed over the cornice (0.125 m in height), while the field observation station is located 400 m upstream of the cornice study site that contains a 220 m plateau ridgeline, which is also not the real wind speed over the cornice, moreover there is a height difference between the field observation station (Gruvefjellet met station, 464 m a.s.l.) and the cornice study site (about 460 m a.s.l., Figure 2 in the paper of Vogel et al. 2012).

Thus, we used the nondimensionalization method to unify the wind speed range between field studies and our laboratory works, using the threshold speed. In our experiments, the cornice accretion starts after the wind speed exceeds the threshold wind speed (3.2 m s$^{-1}$) with enough snow supply, and no cornice formation when the wind speed is below it (e.g., 3.0 m s$^{-1}$). In field observation, Vogel et al. (2012) surmised that cornice accretion proceeded during both entire snow seasons (46h in 2008/2009, 54h in 2009/2010), when wind speeds averaged 12 m s$^{-1}$, with a minimum of at least **10 m s$^{-1}$**, marking the lower limit of the cornice accretion; Eckerstorfer et al. (2012) measured that the initial cornice accretion started along the plateau edge during the first snowfall (10-12, Oct. 2010) with maximum wind speeds of **11 m s$^{-1}$**, which is lower than the observation result of Vogel et al. (2012). Thus, by analyzing the time series of wind speed data from Gruvefjellet meteo station (http://158.39.149.183/Gruvefjellet/index.html), we can conclude that the corresponding averaged wind speed is about $7.37 \pm 0.97$ m s$^{-1}$ (and mean friction velocity is about 0.288 m s$^{-1}$ using $z_0 = 10^{-4}$ m) when the maximum wind speed is in the range of 10.5~11.5 m s$^{-1}$. Considering the harder snow surface in Gruvefjellet (Eckerstorfer et al., 2012), this wind speed value is comparable to the threshold wind speed in previous literature. For the wind in the field is gustier and more turbulent compared to the wind tunnel, the actual threshold wind speed value for cornice growth in the non-dimensional calculation should equal the maximum value of the threshold wind speed measured in the field, according to the study of Li et al. (2020, DOI: 10.1029/2019GL086574). Thus, we chose the maximum wind speed value 11 m s$^{-1}$ as the comparison value with the threshold wind speed in our laboratory experiment where the wind condition is stable and steady. The nondimensional velocity is then defined as: $\tilde{u} = \frac{u_{ref}}{u_{t\_ref}} = \frac{u_*}{u_{*t}}$, in which $u_{ref}$, $u_*$ are the wind speed at the measurement height and friction velocity, respectively, and $u_{t\_ref}$, $u_{*t}$ are the corresponding threshold wind speed at the measurement height and threshold friction velocity, respectively.

To better explain the reason for the cornice formation in drifting snow, we set up a new conceptual model on the basis of granular snow continuously interacting with the wind during transportation, as shown in Section 4 Discussion. We then used the model to estimate the length growth rate in the field and compared the results with the filed observations. The estimated suitable wind speeds are consistent with the observation results from the field, and the estimated length growth rates are comparable with the TLS data.

***Specific comments:***

1. Title – I wonder if the title could be more specific than "environmental conditions" – would wind conditions be more appropriate?

**Response:** Thank you for the suggestion. We will revise the title to "Wind Conditions for Snow Cornice Formation in a Wind Tunnel".

2. Line 16 – please clarify what the percentages refer to, or consider omitting the percentages altogether.

**Response:** The percentages refer to the contribution of different types that causes secondary snow avalanche in the snow seasons 2006-2009 in Longyearbyen. For clarity, we have omitted the percentages in the revised manuscript.

3. Lines 27-31 – consider splitting this long sentence for clarity and readability

**Response:** Thank you for pointing this out, we revised it as shown in lines 38-41: "Indirect evidence was presented by van Herwijnen and Fierz (2014) that snow cornices only grow under the moderate to strong wind, during or soon after the snowfall. The cornice width from observation is in remarkable agreement with the wind drift index calculated by the snow cover model SNOWPACK (Lehning and Fierz, 2008), which indicates that snow transport plays an important role in cornice formation."

4. Lines 36-38 – please revise this sentence, I don't understand what widely accepted hypothesis is referred to here

**Response:** We have deleted this sentence which might lead to misunderstanding: "Particles, which follow the changing wind direction locally as the flow passes the ridge will stick to the growing cornice front at the mountain ridge is a widely accepted hypothesis."

5. Lines 39-40 – which assumptions have no supporting evidence?

**Response:** In here, we mean these assumptions of the vortex and the electric fields lack supporting experimental evidence. In the revised manuscript, we have deleted this sentence.

6. Lines 41-42 – this sentence needs to be revised in lieu of the existence of the Naitou and Kobayashi work. I am unable to read Japanese so cannot specifically comment on the methodological and content overlap between this work and the Naitou and Kobayashi study. I would suggest attempting to determine how your work differs from this previous work and adjusting the intro/results as needed.

**Response:** Thank you very much for the suggestion. Our differences are listed below:

(1) So far, there are few laboratory experiments on cornice formation, except for the work of N&K86. However, due to the language gap, few researchers have noticed this non-peer-reviewed Japanese dissertation. Our work can not only enhance the reliability of N&K86 by reproducing their results but also attract more researchers' attention to the snow cornice topic through full peer review and open discussions, which is still valuable.

(2) We recorded the whole cornice formation by implementing a state-of-art technology: the shadowgraphy method, which allowed us to present more details on snow cornice formation. For example, we quantitively measured the whole growth process of snow cornice. We recorded the time series of cornice growth rates in length and thickness and particle mass flux in the air, etc. These experimental data are fundamental for further conceptual investigation of cornice formation.

(3) We proposed a conceptual model to explain the length and thickness growth of snow cornice based on the mass balance conservation and the experimental results, which have a good consistent with the field observation results in Gruvefjellet, Svalbard, Norway (Vogel et al., 2012; Hancock et al., 2020).

(4) We also tested the factors of the air temperature and grain shape on the snow cornice formation. However, these effects are not dominant as the wind speed. Thus, in this manuscript, we mainly focused on the wind effects and didn't mention these results.

More details can be seen in the revised manuscript. For example, we summarized

their work and pointed out the open scientific questions in lines 30-31: "Naito and Kobayashi (1986) measured the suitable wind speed for cornice formation is between 4 m s$^{-1}$ to 8 m s$^{-1}$, at 1 m above the snow surface in the field and at the center (0.5 m height) in the wind tunnel." and lines 46-53: "There are few laboratory experiments on cornice formation except Naruse (1985) and Naito and Kobayashi (1986). Naito and Kobayashi (1986) carried out experiments both in the wind tunnel and in the field, observing the process of snow cornice. They described the snow cornice formation as a process that snow particles adhere one after another at the leeward edge, in the form of a thin slab of snow elongating leeward, then the slab hangs down under its weight, depositing drifted snow particles on it. However, quantitative descriptions of this process have not been reported. Their results show that the cornice growth under suitable conditions of the air temperature is between -20 °C to 0 °C, the wind speed is between 4 m s$^{-1}$ to 8 m s$^{-1}$, and fresh snow with an irregular dendritic shape. However, further quantitative analysis of experiments has not been carried out."

In section 3.2, we added the comparison results of our collection rates value in lines 165-169:

"A non-dimensional wind speed $\tilde{u} = \frac{u}{u_t}$ is defined here to compare with the experimental results of Naito and Kobayashi (1986). In this definition, $u_t$ is the threshold wind speed which can be considered as the lower limit wind speed value for cornice growth. As is shown in Fig. 5, the mass collection efficiency in both experiments decreases with the increasing wind speed and the corresponding drift rate. Our experimental results are much larger than that in N&K86, which is mainly due to the different wind tunnel sizes."

[Figure]

Fig. 5 Collection efficiency $E$ (in blue) and snow transport rate $Q$ (in red) under different

non-dimensional wind speeds $\tilde{u}$. X represents the distance from the snow particle feeding point to the mass collection pits where the cornice grows. Lines are for ring wind-tunnel experiments, hollow scatters are for N&K86. N&K86 represents the experiment results of (Naito and Kobayashi, 1986).

7. Line 53 – is 7 m s$^{-1}$ the maximum wind speed this device can generate?

**Response:** The maximum wind speed can reach 8 m s$^{-1}$. However, in our experiment, we only use the range of 3 - 7 m s$^{-1}$ for there is no cornice when $U \leq 3$ m s$^{-1}$ and $U \geq 6.5$ m s$^{-1}$.

8. Lines 108-113 – Super cool! Thanks for this.

**Response:** Thanks for your field observation data.

9. Figure 3 caption – I think the cornice length growth rate and cornice thickness growth rate are represented with Xs, not triangles.

**Response:** Thank you for pointing this out, we will correct the caption to "cornice length growth rate (light blue crosses), cornice thickness growth rate (grey crosses)".

10. Line 144 – by crackdown do you mean cornice collapse or failure?

**Response:** Thank you for pointing this out, we will change the word "crackdown" to "collapse" for precise expression in the revision.

11. Line 145-146 – Is this because higher wind speeds form a cornice with a smaller angle?

**Response:** Yes, cornices form at a higher wind speed with a smaller angle.

12. Lines 175-185 – I am struggling to follow these calculations here, which may partially be due to my inexperience with such work. Is it possible to more explicitly define your terms somewhere (e.g. in a table) in the manuscript to help along readers such as myself? Also, to what are you referencing the Leonard et al. (2012) study here?

**Response:** Thanks for your comment, we will add a notation with physical definitions and units of all the symbols in the manuscript.

For Leonard et al. (2012) study, we were referencing the threshold friction velocity. However, we deleted the sentence "where $u_{*t} = 0.25$ m s$^{-1}$ is the threshold friction velocity Leonard et al. (2012)" to avoid misunderstanding.

13. Line 204 – please revise this sentence in lieu of Naitou and Kobayashi.

**Response:** We have deleted the word "first" in this sentence.

14. Lines 205 – 215 – specifically here I think this work would really benefit from explicitly linking your results to field meteorological measurements (e.g. Ut2.8) for increased utility of your results and work.

**Response:** Thanks for this valuable suggestion. We have reorganized it by splitting the linking wind tunnel results to field meteorological measurements as a separate section of the Discussion.

15. Technical corrections:

Line 28 – that snow cornices only grow

Line 135 – there are no more chances for slabs to form on the model edge because of…

Line 137 – gets smaller

Line 202 – the newly formed cornice

**Response:** Thank you for pointing these out. We have revised them in the manuscript.

---

## Referee Report (RR1)

*General comments:*

Thank you again for the opportunity to review the revised version of this manuscript. The current version of the manuscript, in my opinion, represents a considerable improvement in terms of readability and the ease in which it allows the authors' work to be understood. I particularly appreciate the authors' efforts to expand the literature review in the introduction and more explicitly address previous work by Naito and Kabayashi in this version of the manuscript. Additionally, the inclusion of the notation section and the reworking of the results and discussion section – together with Figure 7 and the conceptual model for cornice formation – makes this version considerably more accessible to a broader audience.

This work utilizes repeatable, controlled laboratory experiments to improve our understanding of snow cornice processes. In particular, this work provides a solid theoretical and numerical foundation which can help inform and improve future field studies and serve as a basis for additional modeling efforts. Accordingly, I believe this work will appeal to a relatively broad scientific audience interested in snow processes and also potentially to practitioners working with snow avalanche and cornice hazards.

Certain sections of the work suffer from decreased readability due to language issues. In most cases these issues will be easily resolved with an additional editing session, but in a few cases I found language issues to impede understanding of the scientific concepts being described. I've included these minor issues in the specific comments and have also tried to include some of the more noticeable technical corrections.

*Specific comments:*

Line 8: I think you should define $\bar{u}$ here.

Line 14: Consider briefly specifying how this work can contribute to improved snow avalanche forecasting (maybe via improved understanding of cornice processes which can influence avalanche activity?).

Lines 43-44: I think "daily observations" better describes the temporal issues you allude to here rather than "average observations."

Lines 44: do you mean the compromises necessary to acquire these field observations?

Lines 51-52: The language in this sentence is a bit confusing and should be revised, but from a content perspective do they find fresh snow with an irregular dendritic shape needs to be available for wind transport for cornices to form? It would help the reader to be a bit more specific about "fresh snow with an irregular dendritic shape."

Line 53: mainly occurs through snow redistribution processes?

Line 54-55: Gauer (2001) could reproduce cornice formations numerically due to poorly understood physical formation mechanisms?

Line 85-86: the SSA provided here is for snow stored for a few days up to a week?

Figure 3: Nice!

Line 170-171: Here are you describing that collection efficiency refers to temporary storage of the snow particles under transport? I am struggling to understand this description as it is currently written.

Line 185: Cool! This is a super useful result.

Line 208: I am unsure what "sticking particles at the edge" refers to here?

*Technical corrections:*

Line 3-4: consider "This is particularly true with respect to the wind conditions which favor cornice formation" to make this a complete sentence

Line 23: infrastructure

Lines 28-29: Montagne et al. (1968)

Line 29: measured a wind speed range between 7 and 15 ms$^{-1}$ for cornice formation

Line 30-31: consider "identified wind speeds between 4 to 8 ms$^{-1}$ as suitable for cornice formation at 1 m above…"

Line 37: these discrepancies

Lines 38-39: under moderate to strong winds

Lines 48-50: This sentence should be rewritten as two sentences.

Lines 51-52: maybe "Their results show suitable conditions for cornice growth include air temperature between -20 deg C and 0 deg C, wind speeds between 4 and 8 ms-1, and"

Line 171: This value only reflects

Line 173: remove "the" before Section 3.3

Line 206: a snow cornice

Line 209: can be assumed to be a repeated process.

Line 265: Hancock :)

Line 313: do you mean field data here?

Line 316: remove of from before threshold wind speed

Line 316: field.

Line 320: observational data

Line 321: and measurements of other relevant parameters.

---

## Referee Report (RR2)

The manuscript has been well-revised. The authors referred to similar experiments made by Naito and Kobayashi (1986) then succeeded in quantitatively showing the formation process of snow cornices governed by the balance of growth and erosion rates. This study will provide an important contribution to the observation method of snow cornice and snow cornice-induced avalanche forecasting. I would recommend it for acceptance after the minor points listed below are addressed.

L125
The formulation of Eq. (7) seems not described. More explanation from Eq. (6) to Eq. (7) should be given.

Section 4.2.1
The threshold wind speed ($u_t$) is set as 10 ms$^{-1}$, however, this can give the misleading impression that $u_t$ is too much higher than the wind tunnel experiments.
In the reply to referee #1's comments, the authors estimated the friction velocity and concluded that this wind speed value is comparable to the threshold wind speed in previous literature. This quantification process should be introduced here.

L123
… $\Phi_p(z)$ is the mass concentration calculated by Eq. (1) …
Eq. (5)?

L265
Hancoko et al. (2020)
-> Hancock

L346
Kosugi, K., … Sato, A., and Prevention, D.
Prevention, D. seems not a human name

L364
Naito and Kobayashi
-> Naito, A. and Kobayashi, D.

---

## Author Response (AR2)

**Point-By-Point Reply to Editor's Comments**

Thank you very much for your manuscript revision. As you will see there is variety in opinions from the reviewers. In general the concerns from the previous round have been addressed. There remains an issue that this has parallels with earlier research around wind tunnel experiments. However, this is such an understudied topic that I consider publication of these experiments is worthwhile. Please bear this point in mind and strengthen the discussion around how this manuscript changes the scientific discourse as you address the remaining comments from the reviewers.

**Response**: We would like to thank the editor for taking the time and patience in review process. We sincerely appreciate your effort in giving us valuable comments and encouragement, which helped us in improving this manuscript.

To strengthen the discussion around how this manuscript changes the scientific discourse, we added lines 334-336 in the conclusion: "Overall, this study is a step forward in understanding the mechanism of cornice formation with detailed measurements and controlled environmental conditions. We also present progress in the methodology of observing snow cornice formation. In the future, this may lead to improvements in cornice-fall avalanche predictions". In the meantime, we also added lines 14-17 in the abstract: "Based on the physics of drifting snow, our results provide new insights into snow cornice formation and improve understanding of cornice processes that can influence avalanche activity. The experimental results and the conceptual model can be used in future snow cornice simulation and prediction work for cornice-induced avalanches."

**Point-By-Point Reply to Referee #2's Comments**

*General comments:*
Thank you again for the opportunity to review the revised version of this manuscript. The current version of the manuscript, in my opinion, represents a considerable improvement in terms of readability and the ease in which it allows the authors' work to be understood. I particularly appreciate the authors' efforts to expand the literature review in the introduction and more explicitly address previous work by Naito and Kabayashi in this version of the manuscript. Additionally, the inclusion of the notation section and the reworking of the results and discussion section – together with Figure 7 and the conceptual model for cornice formation – makes this version considerably more accessible to a broader audience.

This work utilizes repeatable, controlled laboratory experiments to improve our understanding of snow cornice processes. In particular, this work provides a solid theoretical and numerical foundation which can help inform and improve future field studies and serve as a basis for additional modeling efforts. Accordingly, I believe this work will appeal to a relatively broad scientific audience interested in snow processes and also potentially to practitioners working with snow avalanche and cornice hazards.
**Response:** We would like to thank you for your endorsement of our work! We appreciate your efforts and expertise that you contribute to improving this manuscript.

Certain sections of the work suffer from decreased readability due to language issues. In most cases these issues will be easily resolved with an additional editing session, but in a few cases I found language issues to impede understanding of the scientific concepts being described. I've included these minor issues in the specific comments and have also tried to include some of the more noticeable technical corrections.
**Response:** Sorry for the language issues in the manuscript. We will pay more careful attention in revising it and make it readable.

*Specific comments*:
Line 8: I think you should define $\bar{u}$ here.
**Response:** To keep this sentence concise, we revised this sentence in lines 7-8 to "The results show that cornices only appear under a moderate wind speed range (1-2 times threshold wind speed)."

Line 14: Consider briefly specifying how this work can contribute to improved snow avalanche forecasting (maybe via improved understanding of cornice processes which can influence avalanche activity?).
**Response:** Thank you for this important comment. We have added this to lines 14-17: "Based on the physics of drifting snow, our results provide new insights based on the physics of drifting snow into snow cornice formation and improve understanding of cornice processes that can influence avalanche activity. The experimental results and

the conceptual model can be used in future snow cornice simulation and prediction work for cornice-induced avalanches."

Lines 43-44: I think "daily observations" better describes the temporal issues you allude to here rather than "average observations."
**Response:** We have revised the sentence in lines 45-47 to: "However, cornices often grow through relatively discrete events in the field (Vogel et al., 2012; van Herwijnen and Fierz, 2014; Naito and Kobayashi, 1986; Hancock et al., 2020). Daily observations therefore only incompletely characterize cornice growth conditions."

Lines 44: do you mean the compromises necessary to acquire these field observations?
**Response:** Yes, I mean some uncontrollable factors for example weather that can affect the continuity of the field observations.

Lines 51-52: The language in this sentence is a bit confusing and should be revised, but from a content perspective do they find fresh snow with an irregular dendritic shape needs to be available for wind transport for cornices to form? It would help the reader to be a bit more specific about "fresh snow with an irregular dendritic shape."
**Response:** Sorry for the misunderstanding we made in this sentence. They also observed that fresh snow in an irregular dendriform shape is more appropriate for the cornice formation than the aged round snow, which might because of their larger contact surface. In order to express more clearly, we revised this sentence in lines 54-55 to: "Their results show suitable conditions for cornice growth include the air temperatures of -20 °C to 0 °C, wind speeds 4 m s$^{-1}$ to 8 m s$^{-1}$, and irregular dendritic-shaped snowflakes with larger contact surface."

Line 53: mainly occurs through snow redistribution processes?
**Response:** I agree with you. It is more proper to use "redistribution" here. We have revised this word.

Line 54-55: Gauer (2001) could reproduce cornice formations numerically due to poorly understood physical formation mechanisms?
**Response:** To avoid misunderstanding, we revised the sentence in lines 56-58 to "Moreover, the locations such as cornice-like deposition at the ridge are well predicted in the numerical simulation using Alpine3D (Lehning et al., 2008) and ARPS (Mott et al., 2010), but the cornice shape cannot be represented."

Line 85-86: the SSA provided here is for snow stored for a few days up to a week?
**Response:** Sorry, in here we made a mistake, we have revised this sentence in lines 87-88 to "The specific surface area (SSA) was about 40-60 mm$^{-1}$ for the snow that was stored a few days up to a week (Schleef, 2014)."

Figure 3: Nice!
**Response:** We thank the reviewer for a positive comment.

Line 170-171: Here are you describing that collection efficiency refers to temporary storage of the snow particles under transport? I am struggling to understand this description as it is currently written.

**Response:** The collection efficiency is the ratio of permanent storage of snow particles to mass flux of airborne particles, and its calculation formula is shown as equation 9. Here, we mean that the cornice growth rate is not only dependent on the collection efficiency, but also determined by the mass flux.

Line 185: Cool! This is a super useful result.

**Response:** We thank the reviewer for a positive comment.

Line 208: I am unsure what "sticking particles at the edge" refers to here?

**Response:** Sorry for the misleading. Here, we mean that the particles stopped at the edge. We revised this sentence in lines 212-213 to: "The first stage can be assumed as a formation of a 1-2 particle diameters thick snow slab composed of particles sticking horizontally at the edge (see Fig. 3a and b). "

***Technical corrections:***

Line 3-4: consider "This is particularly true with respect to the wind conditions which favor cornice formation" to make this a complete sentence

**Response:** Thank you for pointing this out, we have corrected this sentence in lines 3-4 to "This is particularly true with respect to wind conditions which favor cornice formation."

Line 23: infrastructure

**Response:** Thank you for pointing this out, we have corrected the word.

Lines 28-29: Montagne et al. (1968)

**Response:** Thank you for pointing this out, we have corrected the name.

Line 29: measured a wind speed range between 7 and 15 ms-1 for cornice formation

**Response:** Thank you for pointing this out, we have corrected the sentence.

Line 30-31: consider "identified wind speeds between 4 to 8 ms-1 as suitable for cornice formation at 1 m above…"

**Response:** Thank you for pointing this out, we have corrected this sentence in lines 33-34 to "Naito and Kobayashi (1986) identified wind speed between 4 to 8 m $s^{-1}$ as suitable for cornice formation at 1 m above the snow surface in the field and at the center (0.5 m height) in the wind tunnel"

Line 37: these discrepancies

**Response:** Thank you for pointing this out, we have corrected the word.

Lines 38-39: under moderate to strong winds

**Response:** Thank you for pointing this out, we have corrected the word.

Lines 48-50: This sentence should be rewritten as two sentences.

**Response:** Thanks. We have corrected this sentence in lines 52-53 to "They described snow cornice formation as a process in which drifting snow particles adhere one after another at the leeward edge. The formed thin snow slab elongating leeward then hangs down under its weight."

Lines 51-52: maybe "Their results show suitable conditions for cornice growth include air temperature between -20 deg C and 0 deg C, wind speeds between 4 and 8 ms-1, and"

**Response:** Thank you for pointing this out, we have corrected this sentence in lines 54-55 to "Their results show suitable conditions for cornice growth include the air temperatures of -20 to 0 $^{\circ}$C, wind speeds 4 to 8 m s$^{-1}$, and irregular dendritic-shaped snowflakes with larger contact surface."

Line 171: This value only reflects

**Response:** Thanks. We have corrected the word.

Line 173: remove "the" before Section 3.3

**Response:** Thank you for pointing this out, we have removed the word "the" before Section 3.3.

Line 206: a snow cornice

**Response:** Thank you for pointing this out, we have added the word "a" before snow cornice.

Line 209: can be assumed to be a repeated process.

**Response:** Thank you for pointing this out, we have corrected this sentence in lines 214-215 to "The second stage can be assumed to be a repeated process of length growth - thickness growth."

Line 265: Hancock :)

**Response:** We are very sorry to make an incorrect spelling by mistake. We have corrected the name to Hancock.

Line 313: do you mean field data here?

**Response:** Yes, for better illustration here, we revised this sentence in lines 325-327 to "Based on the field observation data, such as roughness length, the threshold wind speed, and the local surface snow conditions, this model can be applied to field conditions to predict the cornice length growth rates and the suitable wind speed range."

Line 316: remove of from before threshold wind speed

**Response:** Thank you for pointing this out, we have revised this sentence.

Line 316: field.
**Response:** Thank you for pointing this out, we have revised this word.

Line 320: observational data
**Response:** Thank you for pointing this out, we have revised this word.

Line 321: and measurements of other relevant parameter
**Response:** Thank you for pointing this out, we have revised this sentence.

**Point-By-Point Reply to Referee #3's Comments**

In this paper, the authors reproduce the development process of snow cornice by wind tunnel experiments in cold room, then constructed a conceptual model using the experiment results. They also applied their conceptual model to actual field observations to verify the practicality of the conceptual model.

As they showed in their paper, although snow cornice is one of the factors that cause avalanches, our understanding of its development process is still insufficient. Therefore, snow cornice role in the avalanche is not sufficiently taken into account in avalanche prediction. For this point, their study may contribute to the avalanche prediction thorough the modeling of the developmental process of snow cornice.
**Response:** We thank the reviewer for a positive view on the importance of our study.

On the other hand, as noted in their paper, wind tunnel experiments and field observations of snow cornice have been conducted as previous studies, although they are few in number. Previous studies have reported that snow cornices develop under moderate wind conditions.

The results of their experiments show similar results. In this sense, the present study can be positioned as an example of wind tunnel experiments on snow cornice development, and it must be said that the results of the experiments themselves have little scientific impact.
**Response:** We agree with you on the point that previous studies have reported that snow cornices develop under moderate wind conditions which had been addressed in the Introduction section in last round revision. Our experiments agree well with the previous research on this point, as a basement results. The main contribution in this paper is that we explained this phenomenon from physical mechanism. Moreover, we developed a concept model that can be used as prediction model for the field. Therefore, we still cannot accept the point that this study just has little scientific impact because it shows the similar results as before.

Their advantage should be that they made more detailed measurements, but it is not clear how their results play a significant role in the interpretation of what has already been reported. For example, they only mentioned that the difference between their result and previous one is caused by wind tunnel size difference, they did not discuss concretely from the scientific view. Therefore, under the present, it is difficult to find the scientific role of their detailed measurements to develop understanding snow cornice development. The comparisons between their conceptual model calculations and actual field observations are only a qualitative comparison due to the limitations of the conceptual model.
**Response:** Thanks. In the calculation of collection efficiency (Eqn. 9), we replaced the particle density by using the cornice density. As the revised results shown in the Figure

5, our experiment results are in the same order of magnitude as that in N&K86's study.

[Figure]

Figure. 5 Collection efficiency $E$ (in blue) and snow transport rate $Q$ (in red) under different non-dimensional wind speeds $\tilde{u}$. $X$ represents the distance from the snow particle feeding point to the mass collection pits where the cornice grows. Lines are for ring wind-tunnel experiments, and hollow scatters are for N&K86. N&K86 represents the experiment results of (Naito1986).

Our measurements do provide more details than the previous study of N&K86. In N&K86's study, only very little raw data and environmental condition are reported. Therefore, we could barely do a deeper comparison between the two works. For readers clear this fact, we revised the sentences in lines 172-174: "Our measured values for the collection efficiency are in the same order of magnitude as in N&K86. Due to the limited data and unpublished details in the study of N&K86, we could not make a deeper comparison."

For these reasons, it must be said that, at this stage, the scientific value of their study is insufficient for publication in the TC.
There is no doubt that a lot of hard work went into this study. The experimental results have sufficient credibility. The manuscript itself is also well described, including the description of the methodology.
Before resubmitting the manuscript, I strongly recommend that the authors reorganize the manuscript to clarify what new scientific findings they have derived from this study and how their finding contribute to advance the scientific topic of the developmental process of snow cornice.
Then they should concretely clarify their scientific contribution in the manuscript.
**Response:** We thank the reviewer for the approval of the work volume, credibility, and the description. To make concretely clarify our scientific contribution, we added lines 334-336 in the conclusion: "Overall, this study is a step forward in understanding the mechanism of cornice formation with detailed measurements and controlled environmental conditions. We also present progress in the methodology of observing

snow cornice formation. In the future, this may lead to improvements in cornice-fall avalanche predictions". In the meantime, we also added lines 14-17 in the abstract: "Based on the physics of drifting snow, our results provide new insights into snow cornice formation and improve understanding of cornice processes that can influence avalanche activity. The experimental results and the conceptual model can be used in future snow cornice simulation and prediction work for cornice-induced avalanches."

**Point-By-Point Reply to Referee #4's Comments**

The manuscript has been well-revised. The authors referred to similar experiments made by Naito and Kobayashi (1986) then succeeded in quantitatively showing the formation process of snow cornices governed by the balance of growth and erosion rates. This study will provide an important contribution to the observation method of snow cornice and snow cornice-induced avalanche forecasting. I would recommend it for acceptance after the minor points listed below are addressed.

**Response:** We thank the reviewer for pointing out many important comments, which are very helpful in improving our manuscript.

L125

The formulation of Eq. (7) seems not described. More explanation from Eq. (6) to Eq. (7) should be given.

**Response:** Thank you for this important comment. It is our mistake that didn't describe enough on the transfer from Eq. (6) to Eq. (7). We have added the explanation in the new version manuscript in line 125: "The transport mass flux profile can be described by an exponential law (Nishimura and Nemoto, 2005; Sugiura et al., 1998):"

Section 4.2.1

The threshold wind speed (ut) is set as 10 ms-1, however, this can give the misleading impression that ut is too much higher than the wind tunnel experiments.

In the reply to referee #1's comments, the authors estimated the friction velocity and concluded that this wind speed value is comparable to the threshold wind speed in previous literature. This quantification process should be introduced here.

**Response:** We agree with you. To avoid this misleading impression, we added this quantification process in lines 244-249: "It should be noted here that by analyzing the time series of wind speed data from Gruvefjellet meteo station (2022), the corresponding averaged wind speed is found to be $7.37 \pm 0.97$ m s$^{-1}$ when the maximum wind speed is in the range of 10.5 to 11.5 m s$^{-1}$. Thus, the friction velocity is 0.29 m s$^{-1}$ assuming a roughness length $z_0 = 10^{-4}$ m. This value is comparable to the threshold wind speed in previous research (Sugiura,1998; Jdoorschot,2004; Clifton, 2006), considering the harder snow surface in Gruvefjellet (Eckerstorfer2013)."

L123

$\cdots$ $\Phi p(z)$ is the mass concentration calculated by Eq. (1) $\dots$

Eq. (5)?

**Response:** Sorry for this mistake we made, and we have corrected it to Eq. (5).

L265

Hancoko et al. (2020)

-> Hancock

**Response:** Sorry for this mistake we made, and we have revised the name.

L346
Kosugi, K., … Sato, A., and Prevention, D.
Prevention, D. seems not a human name
**Response:** Thank you for pointing this out, we have revised the citation here.

L364
Naito and Kobayashi
-> Naito, A. and Kobayashi, D.
**Response:** Thank you for pointing this out, we have revised the citation here.

---

## Author Response (AR3)

**Point-By-Point Reply to Editor's Comments**

Thank you for the changes you have made to this manuscript. This should now go forward for publishing after the following technical corrections have been made:

Response: We would like to thank you for handling and reviewing our manuscript. Thank you for your helpful comments and patience in this review process. We have addressed all the corrections in the final version.

1. Line 73 - is this enhanced cohesion compared with colder temperatures?

Response: Thanks for pointing this out. We have revised this sentence in lines 72 to 73 as: "At this temperature, the cohesion of snow particles is significantly enhanced compared with colder temperatures (Tobias et al., 2022)."

2. Line 191 - average net deposition rate is a confusing term - is this the same as the growth rate (equation 4), in which case perhaps include 'i.e. growth rate'. If it is not the same, please clarify.

Response: Yes, it is indeed the same meaning, and we have revised this sentence in lines 190 to 191 to: "The averaged cornice length growth rate $\overline{l_g}$ (equal to $\overline{l_d} - \overline{l_e}$) reaches its maximum when the wind speed is approximately 40% higher than the threshold wind speed."

3. Line 195. The length erosion rate is approximately 30% lower than the thickness erosion rate (mathematically this is not the same as the thickness erosion rate being 30% higher).

Response: Thanks. It is indeed a misrepresentation. We have revised this sentence in lines 193 to 194 to: "The erosion in length takes place later than in thickness, and the averaged thickness erosion rate is always approximately 30% higher than the length erosion rate ($h_e = 1.3\, l_e$)."

4. Line 329. Add 'for this site' to the end of the sentence to link it to Gruvefjellet rather than the laboratory measurements.

Response: Thanks for this comment. We have revised this sentence in lines 327 to 329 to: "It is found that the most favorable wind condition for cornice growth is approximately 30% higher than the local threshold wind speed for this site."

I would like to thank the authors and all reviewers for their work and inputs to this paper, which are publicly available. There are a range of opinions given earlier historic laboratory measurements, with two positive reviews and a third noting the potential contribution to avalanche applications. None identify any technical flaws. Taking all things into account I consider this worthy of publication as it models cornice formation in a way that can be used in snow evolution and avalanche models, links to other field measurements and this is generally an understudied area.

Response: We appreciate the editor and reviewers in taking the time in reviewing and helping us improving the manuscript.